# BREGMAN GRADIENT POLICY OPTIMIZATION

**Feihu Huang**[*][†], **Shangqian Gao**[*], **Heng Huang**[†]
Department of Electrical and Computer Engineering
University of Pittsburgh
Pittsburgh, PA 15261, USA
`huangfeihu2018@gmail.com, shg84@pitt.edu, heng.huang@pitt.edu`

## ABSTRACT

In the paper, we design a novel Bregman gradient policy optimization framework for reinforcement learning based on Bregman divergences and momentum techniques. Specifically, we propose a Bregman gradient policy optimization (BGPO) algorithm based on the basic momentum technique and mirror descent iteration. Meanwhile, we further propose an accelerated Bregman gradient policy optimization (VR-BGPO) algorithm based on the variance reduced technique. Moreover, we provide a convergence analysis framework for our Bregman gradient policy optimization under the nonconvex setting. We prove that our BGPO achieves a sample complexity of $O(\epsilon^{-4})$ for finding $\epsilon$-stationary policy only requiring one trajectory at each iteration, and our VR-BGPO reaches the best known sample complexity of $O(\epsilon^{-3})$, which also only requires one trajectory at each iteration. In particular, by using different Bregman divergences, our BGPO framework unifies many existing policy optimization algorithms such as the existing (variance reduced) policy gradient algorithms such as natural policy gradient algorithm. Extensive experimental results on multiple reinforcement learning tasks demonstrate the efficiency of our new algorithms.

## 1 INTRODUCTION

Policy Gradient (PG) methods are a class of popular policy optimization methods for Reinforcement Learning (RL), and have achieved significant successes in many challenging applications (Li, 2017) such as robot manipulation (Deisenroth et al., 2013), the Go game (Silver et al., 2017) and autonomous driving (Shalev-Shwartz et al., 2016). In general, PG methods directly search for the optimal policy by maximizing the expected total reward of Markov Decision Processes (MDPs) involved in RL, where an agent takes action dictated by a policy in an unknown dynamic environment over a sequence of time steps. Since the PGs are generally estimated by Monte-Carlo sampling, such vanilla PG methods usually suffer from very high variances resulted in slow convergence rate and destabilization. Thus, recently many fast PG methods have been proposed to reduce variances in vanilla stochastic PGs. For example, Sutton et al. (2000) introduced a baseline to reduce variances of the stochastic PG. Konda & Tsitsiklis (2000) proposed an efficient actor-critic algorithm by estimating the value function to reduce effects of large variances. (Schulman et al., 2016) proposed the generalized advantage estimation (GAE) to control both the bias and variance in policy gradient. More recently, some faster variance-reduced PG methods (Papini et al., 2018; Xu et al., 2019a; Shen et al., 2019; Liu et al., 2020; Huang et al., 2020) have been developed based on the variance-reduction techniques in stochastic optimization.

Alternatively, some successful PG algorithms (Schulman et al., 2015; 2017) improve convergence rate and robustness of vanilla PG methods by using some penalties such as Kullback-Leibler (KL) divergence penalty. For example, trust-region policy optimization (TRPO) (Schulman et al., 2015) ensures that the new selected policy is near to the old one by using KL-divergence constraint, while proximal policy optimization (PPO) (Schulman et al., 2017) clips the weighted likelihood ratio to implicitly reach this goal. Subsequently, Shani et al. (2020) have analyzed the global convergence properties of TRPO in tabular RL based on the convex mirror descent algorithm. Liu et al.

---

[*]Feihu and Shangqian contributed equally.
[†]Corresponding Authors.

Table 1: **Sample complexities** of the representative PG algorithms based on mirror descent algorithm for finding an $\epsilon$-stationary policy of the **nonconcave** performance function. Although Liu et al. (2019); Shani et al. (2020) have provided the global convergence of TRPO and PPO under some specific policies based on convex mirror descent, they still obtain a stationary point of nonconcave performance function. **Note that** our convergence analysis does not rely any specific policies.

| Algorithm | Reference | Complexity | Batch Size |
|---|---|---|---|
| TRPO | Shani et al. (2020) | $O(\epsilon^{-4})$ | $O(\epsilon^{-2})$ |
| Regularized TRPO | Shani et al. (2020) | $O(\epsilon^{-3})$ | $O(\epsilon^{-2})$ |
| TRPO/PPO | Liu et al. (2019) | $O(\epsilon^{-8})$ | $O(\epsilon^{-6})$ |
| VRMPO | Yang et al. (2019) | $O(\epsilon^{-3})$ | $O(\epsilon^{-2})$ |
| MDPO | Tomar et al. (2020) | Unknown | Unknown |
| BGPO | Ours | $O(\epsilon^{-4})$ | $O(1)$ |
| VR-BGPO | Ours | $O(\epsilon^{-3})$ | $O(1)$ |

(2019) have also studied the global convergence properties of PPO and TRPO equipped with over-parametrized neural networks based on mirror descent iterations. At the same time, Yang et al. (2019) tried to propose the PG methods based on the mirror descent algorithm. More recently, mirror descent policy optimization (MDPO) (Tomar et al., 2020) iteratively updates the policy beyond the tabular RL by approximately solving a trust region problem based on convex mirror descent algorithm. In addition, Agarwal et al. (2019); Cen et al. (2020) have studied the natural PG methods for regularized RL. However, Agarwal et al. (2019) mainly focuses on tabular policy and log-linear, neural policy classes. Cen et al. (2020) mainly focuses on softmax policy class.

Although these specific PG methods based on mirror descent iteration have been recently studied, which are scattered in empirical and theoretical aspects respectively, it lacks a universal framework for these PG methods without relying on some specific RL tasks. In particular, there still does not exist the convergence analysis of PG methods based on the mirror descent algorithm under the nonconvex setting. Since mirror descent iteration adjusts gradient updates to fit problem geometry, and is useful in regularized RL (Geist et al., 2019), there exists an important problem to be addressed:

> *Could we design a universal policy optimization framework based on the mirror descent algorithm, and provide its convergence guarantee under the non-convex setting ?*

In the paper, we firmly answer the above challenging question with positive solutions and propose an efficient Bregman gradient policy optimization framework based on Bregman divergences and momentum techniques. In particular, we provide a convergence analysis framework of the PG methods based on mirror descent iteration under the nonconvex setting. In summary, our main contributions are provided as follows:

    a) We propose an effective Bregman gradient policy optimization (BGPO) algorithm based on the basic momentum technique, which achieves the sample complexity of $O(\epsilon^{-4})$ for finding $\epsilon$-stationary policy only requiring one trajectory at each iteration.

    b) We propose an accelerated Bregman gradient policy optimization (VR-BGPO) algorithm based on the variance-reduced technique of STORM (Cutkosky & Orabona, 2019). Moreover, we prove that the VR-BGPO reaches the best known sample complexity of $O(\epsilon^{-3})$.

    c) We design a unified policy optimization framework based on mirror descent iteration and momentum techniques, and provide its convergence analysis under nonconvex setting.

In Table 1 shows that sample complexities of the representative PG algorithms based on mirror descent algorithm. Shani et al. (2020); Liu et al. (2019) have established global convergence of a mirror descent variant of PG under some pre-specified setting such as over-parameterized networks (Liu et al., 2019) by exploiting these specific problems' hidden convex nature. Without these special structures, global convergence of these methods cannot be achieved. However, our framework does not rely on any specific policy classes, and our convergence analysis only builds on the general nonconvex setting. Thus, we only prove that our methods convergence to stationary points.

Geist et al. (2019); Jin & Sidford (2020); Lan (2021); Zhan et al. (2021) studied a general theory of regularized MDPs based on **policy space** such as a discrete probability space that generally is discontinuous. Since both the state and action spaces $\mathcal{S}$ and $\mathcal{A}$ generally are very large in practice, the **policy space** is large. While our methods build on **policy' parameter space** that is generally continuous Euclidean space and relatively small. Clearly, our methods and theoretical results are more practical than the results in (Geist et al., 2019; Jin & Sidford, 2020; Lan, 2021; Zhan et al., 2021). (Tomar et al., 2020) also proposes mirror descent PG framework based on policy parameter

space, but it does not provide any theoretical results and only focuses on Bregman divergence taking form of KL divergence. While our framework can collaborate with any Bregman divergence forms.

## 2 RELATED WORKS

In this section, we review some related works about mirror descent-based algorithms in RL and variance-reduced PG methods, respectively.

### 2.1 MIRROR DESCENT ALGORITHM IN RL

Due to easily deal with the regularization terms, mirror descent (a.k.a., Bregman gradient) algorithm (Censor & Zenios, 1992; Beck & Teboulle, 2003) has shown significant successes in regularized RL, which is first proposed in (Censor & Zenios, 1992) based on Bregman distance (divergence) (Bregman, 1967; Censor & Lent, 1981). For example, Neu et al. (2017) have shown both the dynamic policy programming (Azar et al., 2012) and TRPO (Schulman et al., 2015) algorithms are approximate variants of mirror descent algorithm. Subsequently, Geist et al. (2019) have introduced a general theory of regularized MDPs based on the convex mirror descent algorithm. More recently, Liu et al. (2019) have studied the global convergence properties of PPO and TRPO equipped with overparametrized neural networks based on mirror descent iterations. At the same time, Shani et al. (2020) have analyzed the global convergence properties of TRPO in tabular policy based on the convex mirror descent algorithm. Wang et al. (2019) have proposed divergence augmented policy optimization for off-policy learning based on mirror descent algorithm. MDPO (Tomar et al., 2020) iteratively updates the policy beyond the tabular RL by approximately solving a trust region problem based on convex mirror descent algorithm.

### 2.2 (VARIANCE-REDUCED) PG METHODS

PG methods have been widely studied due to their stability and incremental nature in policy optimization. For example, the global convergence properties of vanilla policy gradient method in infinite-horizon MDPs have been recently studied in (Zhang et al., 2019). Subsequently, Zhang et al. (2020) have studied asymptotically global convergence properties of the REINFORCE (Williams, 1992), whose policy gradient is approximated by using a single trajectory or a fixed size mini-batch of trajectories under soft-max parametrization and log-barrier regularization. To accelerate these vanilla PG methods, some faster variance-reduced PG methods have been proposed based on the variance-reduction techniques of SVRG (Johnson & Zhang, 2013), SPIDER (Fang et al., 2018) and STORM (Cutkosky & Orabona, 2019) in stochastic optimization. For example, fast SVRPG (Papini et al., 2018; Xu et al., 2019a) algorithm have been proposed based on SVRG. Fast HAPG (Shen et al., 2019) and SRVR-PG (Xu et al., 2019a) algorithms have been presented by using SPIDER technique. Subsequently, the momentum-based PG methods, i.e., ProxHSPGA (Pham et al., 2020) and IS-MBPG (Huang et al., 2020), have been developed based on variance-reduced technique of STORM/Hybrid-SGD (Cutkosky & Orabona, 2019; Tran-Dinh et al., 2019). More recently, (Ding et al., 2021) studied the global convergence of momentum-based policy gradient methods. (Zhang et al., 2021) proposed a truncated stochastic incremental variance-reduced policy gradient (TSIVR-PG) method to relieve the uncheckable importance weight assumption in above variance-reduced PG methods and provided the global convergence of the TSIVR-PG under overparameterizaiton of policy assumption.

## 3 PRELIMINARIES

In the section, we will review some preliminaries of Markov decision process and policy gradients.

### 3.1 NOTATIONS

Let $[n] = \{1, 2, \cdots, n\}$ for all $n \in \mathbb{N}_+$. For a vector $x \in \mathbb{R}^d$, let $\|x\|$ denote the $\ell_2$ norm of $x$, and $\|x\|_p = \left( \sum_{i=1}^d |x_i|^p \right)^{1/p}$ $(p \geq 1)$ denotes the $p$-norm of $x$. For two sequences $\{a_k\}$ and $\{b_k\}$, we denote $a_k = O(b_k)$ if $a_k \leq C b_k$ for some constant $C > 0$. $\mathbb{E}[X]$ and $\mathbb{V}[X]$ denote the expectation and variance of random variable $X$, respectively.

### 3.2 MARKOV DECISION PROCESS

Reinforcement learning generally involves a discrete time discounted Markov Decision Process (MDP) defined by a tuple $\{\mathcal{S}, \mathcal{A}, \mathbb{P}, r, \gamma, \rho_0\}$. $\mathcal{S}$ and $\mathcal{A}$ denote the state and action spaces of the

agent, respectively. $\mathbb{P}(s'|s,a) : \mathcal{S} \times \mathcal{A} \to \triangle(\mathcal{S})$ is the Markov kernel that determines the transition probability from the state $s$ to $s'$ under taking an action $a \in \mathcal{A}$. $r(s,a) : \mathcal{S} \times \mathcal{A} \to [-R, R]$ $(R > 0)$ is the reward function of $s$ and $a$, and $\rho_0 = p(s_0)$ denotes the initial state distribution. $\gamma \in (0, 1)$ is the discount factor. Let $\pi : \mathcal{S} \to \triangle(\mathcal{A})$ be a stationary policy, where $\triangle(\mathcal{A})$ is the set of probability distributions on $\mathcal{A}$.

Given the current state $s_t \in \mathcal{S}$, the agent executes an action $a_t \in \mathcal{A}$ following a conditional probability distribution $\pi(a_t|s_t)$, and then the agent obtains a reward $r_t = r(s_t, a_t)$. At each time $t$, we can define the state-action value function $Q^\pi(s_t, a_t)$ and state value function $V^\pi(s_t)$ as follows:

$$Q^\pi(s_t, a_t) = \mathbb{E}_{s_{t+1}, a_{t+1}, \dots}\Big[\sum_{l=0}^{\infty} \gamma^l r_{t+l}\Big], \ V^\pi(s_t) = \mathbb{E}_{a_t, s_{t+1}, \dots}\Big[\sum_{l=0}^{\infty} \gamma^l r_{t+l}\Big]. \tag{1}$$

We also define the advantage function $A^\pi(s_t, a_t) = Q^\pi(s_t, a_t) - V^\pi(s_t)$. The goal of the agent is to find the optimal policy by maximizing the expected discounted reward

$$\max_{\pi} J(\pi) := \mathbb{E}_{s_0 \sim \rho_0}[V^\pi(s_0)]. \tag{2}$$

Given a time horizon $H$, the agent collects a trajectory $\tau = \{s_t, a_t\}_{t=0}^{H-1}$ under any stationary policy. Then the agent obtains a cumulative discounted reward $r(\tau) = \sum_{t=0}^{H-1} \gamma^t r(s_t, a_t)$. Since the state and action spaces $\mathcal{S}$ and $\mathcal{A}$ are generally very large, directly solving the problem (2) is difficult. Thus, we let the policy $\pi$ be parametrized as $\pi_\theta$ for the parameter $\theta \in \Theta \subseteq \mathbb{R}^d$. Given the initial distribution $\rho_0 = p(s_0)$, the probability distribution over trajectory $\tau$ can be obtained

$$p(\tau|\theta) = p(s_0) \prod_{t=0}^{H-1} \mathbb{P}(s_{t+1}|s_t, a_t)\pi_\theta(a_t|s_t). \tag{3}$$

Thus, the problem (2) will be equivalent to maximize the expected discounted trajectory reward:

$$\max_{\theta \in \Theta} J(\theta) := \mathbb{E}_{\tau \sim p(\tau|\theta)}[r(\tau)]. \tag{4}$$

In fact, the above objective function $J(\theta)$ has a truncation error of $O(\frac{\gamma^H}{1-\gamma})$ compared to the original infinite-horizon MDP.

### 3.3 POLICY GRADIENTS

The policy gradient methods (Williams, 1992; Sutton et al., 2000) are a class of effective policy-based methods to solve the above RL problem (4). Specifically, the gradient of $J(\theta)$ with respect to $\theta$ is given as follows:

$$\nabla J(\theta) = \mathbb{E}_{\tau \sim p(\tau|\theta)}\big[\nabla \log\big(p(\tau|\theta)\big)r(\tau)\big]. \tag{5}$$

Given a mini-batch trajectories $\mathcal{B} = \{\tau_i\}_{i=1}^n$ sampled from the distribution $p(\tau|\theta)$, the standard stochastic policy gradient ascent update at $(k + 1)$-th step, defined as

$$\theta_{k+1} = \theta_k + \eta \nabla J_{\mathcal{B}}(\theta_k), \tag{6}$$

where $\eta > 0$ is learning rate, and $\nabla J_{\mathcal{B}}(\theta_k) = \frac{1}{n}\sum_{i=1}^n g(\tau_i|\theta_k)$ is stochastic policy gradient. Given $H = O(\frac{1}{1-\gamma})$ as in (Zhang et al., 2019; Shani et al., 2020), $g(\tau|\theta)$ is the unbiased stochastic policy gradient of $J(\theta)$, $i.e.$, $\mathbb{E}[g(\tau|\theta)] = \nabla J(\theta)$, where

$$g(\tau|\theta) = \Big(\sum_{t=0}^{H-1} \nabla_\theta \log \pi_\theta(a_t, s_t)\Big)\Big(\sum_{t=0}^{H-1} \gamma^t r(s_t, a_t)\Big). \tag{7}$$

Based on the gradient estimator in (7), we can obtain the existing well-known policy gradient estimators such as REINFORCE (Williams, 1992), policy gradient theorem (PGT (Sutton et al., 2000)). Specifically, the REINFORCE obtains a policy gradient estimator by adding a baseline $b$, defined as

$$g(\tau|\theta) = \Big(\sum_{t=0}^{H-1} \nabla_\theta \log \pi_\theta(a_t, s_t)\Big)\Big(\sum_{t=0}^{H-1} \gamma^t r(s_t, a_t) - b_t\Big).$$

The PGT is a version of the REINFORCE, defined as

$$g(\tau|\theta) = \sum_{t=0}^{H-1} \sum_{j=t}^{H-1} \big(\gamma^j r(s_j, a_j) - b_j\big)\nabla_\theta \log \pi_\theta(a_t, s_t).$$

---

**Algorithm 1** BGPO Algorithm

---

1: **Input:** Total iteration $K$, tuning parameters $\{\lambda, b, m, c\}$ and mirror mappings $\{\psi_k\}_{k=1}^K$ are $\nu$-strongly convex functions;
2: **Initialize:** $\theta_1 \in \Theta$, and sample a trajectory $\tau_1$ from $p(\tau|\theta_1)$, and compute $u_1 = -g(\tau_1|\theta_1)$;
3: **for** $k = 1, 2, \ldots, K$ **do**
4: $\quad$ Update $\tilde{\theta}_{k+1} = \arg\min_{\theta \in \Theta} \{\langle u_k, \theta \rangle + \frac{1}{\lambda} D_{\psi_k}(\theta, \theta_k)\}$;
5: $\quad$ Update $\theta_{k+1} = \theta_k + \eta_k(\tilde{\theta}_{k+1} - \theta_k)$ with $\eta_k = \frac{b}{(m+k)^{1/2}}$;
6: $\quad$ Sample a trajectory $\tau_{k+1}$ from $p(\tau|\theta_{k+1})$, and compute $u_{k+1} = -\beta_{k+1}g(\tau_{k+1}|\theta_{k+1}) + (1 - \beta_{k+1})u_k$ with $\beta_{k+1} = c\eta_k$;
7: **end for**
8: **Output:** $\theta_\zeta$ chosen uniformly random from $\{\theta_k\}_{k=1}^K$.

---

## 4 BREGMAN GRADIENT POLICY OPTIMIZATION

In this section, we propose a novel Bregman gradient policy optimization framework based on Bregman divergences and momentum techniques. We first let $f(\theta) = -J(\theta)$, the goal of policy-based RL is to solve the problem: $\max_{\theta \in \Theta} J(\theta) \iff \min_{\theta \in \Theta} f(\theta)$, so we have $\nabla f(\theta) = -\nabla J(\theta)$.

Assume $\psi(x)$ is a continuously-differentiable and $\nu$-strongly convex function, i.e., $\langle x - y, \nabla\psi(x) - \nabla\psi(y)\rangle \geq \nu\|x - y\|$, $\nu > 0$, we define a Bregman distance:

$$D_\psi(y, x) = \psi(y) - \psi(x) - \langle\nabla\psi(x), y - x\rangle, \ \forall x, y \in \mathbb{R}^d \tag{8}$$

Then given a function $h(x)$ defined on a closed convex set $\mathcal{X}$, we define a proximal operator (a.k.a., mirror descent):

$$\mathcal{P}_{\lambda, h}^\psi(x) = \arg\min_{y \in \mathcal{X}} \{h(y) + \frac{1}{\lambda}D_\psi(y, x)\}, \tag{9}$$

where $\lambda > 0$. Based on this proximal operator $\mathcal{P}_{\lambda, h}^\psi$ as in (Ghadimi et al., 2016; Zhang & He, 2018), we can define a Bregman gradient of function $h(x)$ as follows:

$$\mathcal{B}_{\lambda, h}^\psi(x) = \frac{1}{\lambda}(x - \mathcal{P}_{\lambda, h}^\psi(x)). \tag{10}$$

If $\psi(x) = \frac{1}{2}\|x\|^2$ and $\mathcal{X} = \mathbb{R}^d$, $x^*$ is a stationary point of $h(x)$ if and only if $\mathcal{B}_{\lambda, h}^\psi(x^*) = \nabla h(x^*) = 0$. Thus, this Bregman gradient can be regarded as a generalized gradient.

### 4.1 BGPO ALGORITHM

In the subsection, we propose a Bregman gradient policy optimization (BGPO) algorithm based on the basic momentum technique. The pseudo code of BGPO Algorithm is provided in Algorithm 1.

In Algorithm 1, the step 4 uses the stochastic Bregman gradient descent (a.k.a., stochastic mirror descent) to update the parameter $\theta$. Let $h(\theta) = \langle\theta, u_k\rangle$ be the first-order approximation of function $f(\theta)$ at $\theta_k$, where $u_k$ is an approximated gradient of function $f(\theta)$ at $\theta_k$. By the step 4 of Algorithm 1 and the above equality (10), we have

$$\mathcal{B}_{\lambda, h}^{\psi_k}(\theta_k) = \frac{1}{\lambda}(\theta_k - \tilde{\theta}_{k+1}), \tag{11}$$

where $\lambda > 0$. Then by the step 5 of Algorithm 1, we have

$$\theta_{k+1} = \theta_k - \lambda\eta_k\mathcal{B}_{\lambda, h}^{\psi_k}(\theta_k), \tag{12}$$

where $0 < \eta_k \leq 1$. Due to the convexity of set $\Theta \subseteq \mathbb{R}^d$ and $\theta_1 \in \Theta$, we choose the parameter $\eta_k \in (0, 1]$ to ensure the updated sequence $\{\theta_k\}_{k=1}^K$ in $\Theta$.

In fact, our BGPO algorithm unifies many popular policy optimization algorithms. When the mirror mappings $\psi_k(\theta) = \frac{1}{2}\|\theta\|^2$ for $\forall k \geq 1$, the update (12) will be equivalent to a classic policy gradient iteration. Then our BGPO algorithm will become a momentum version of the policy gradient algorithms (Sutton et al., 2000; Zhang et al., 2019). Given $\psi_k(\theta) = \frac{1}{2}\|\theta\|^2$ and $\beta_k = 1$, i.e., $u_k = -g(\tau_k|\theta_k)$, we have $\mathcal{B}_{\lambda, h}^{\psi_k}(\theta_k) = -g(\tau_k|\theta_k)$ and

$$\theta_{k+1} = \theta_k + \lambda\eta_k g(\tau_k|\theta_k). \tag{13}$$

---

**Algorithm 2** VR-BGPO Algorithm

---

1: **Input:** Total iteration $K$, tuning parameters $\{\lambda, b, m, c\}$ and mirror mappings $\{\psi_k\}_{k=1}^K$ are $\nu$-strongly convex functions;
2: **Initialize:** $\theta_1 \in \Theta$, and sample a trajectory $\tau_1$ from $p(\tau|\theta_1)$, and compute $u_1 = -g(\tau_1|\theta_1)$;
3: **for** $k = 1, 2, \ldots, K$ **do**
4:    Update $\tilde{\theta}_{k+1} = \arg\min_{\theta \in \Theta} \{\langle u_k, \theta \rangle + \frac{1}{\lambda} D_{\psi_k}(\theta, \theta_k)\}$;
5:    Update $\theta_{k+1} = \theta_k + \eta_k(\tilde{\theta}_{k+1} - \theta_k)$ with $\eta_k = \frac{b}{(m+k)^{1/3}}$;
6:    Sample a trajectory $\tau_{k+1}$ from $p(\tau|\theta_{k+1})$, and compute $u_{k+1} = -\beta_{k+1}g(\tau_{k+1}|\theta_{k+1}) + (1 - \beta_{k+1})[u_k - g(\tau_{k+1}|\theta_{k+1}) + w(\tau_{k+1}|\theta_k, \theta_{k+1})g(\tau_{k+1}|\theta_k)]$ with $\beta_{k+1} = c\eta_k^2$;
7: **end for**
8: **Output:** $\theta_\zeta$ chosen uniformly random from $\{\theta_k\}_{k=1}^K$.

---

When the mirror mappings $\psi_k(\theta) = \frac{1}{2}\theta^T F(\theta_k)\theta$ with $F(\theta_k) = \mathbb{E}[\nabla_\theta \pi_{\theta_k}(s,a)(\nabla_\theta \pi_{\theta_k}(s,a))^T]$, the update (12) will be equivalent to a natural policy gradient iteration. Then our BGPO will become a momentum version of natural policy gradient algorithms (Kakade, 2001; Liu et al., 2020). Given $\psi_k(\theta) = \frac{1}{2}\theta^T F(\theta_k)\theta$, $\beta_k = 1$, i.e., $u_k = -g(\tau_k|\theta_k)$, we have $\mathcal{B}_{\lambda,h}^{\psi_k}(\theta_k) = -F(\theta_k)^+ g(\tau_k|\theta_k)$ and

$$\theta_{k+1} = \theta_k + \lambda\eta_k F(\theta_k)^+ g(\tau_k|\theta_k), \tag{14}$$

where $F(\theta_k)^+$ denotes the Moore-Penrose pseudoinverse of the Fisher information matrix $F(\theta_k)$. When given the mirror mapping $\psi_k(\theta) = \sum_{s \in \mathcal{S}} \pi_\theta(s) \log(\pi_\theta(s))$, i.e., Boltzmann-Shannon entropy function (Shannon, 1948) and $\Theta = \{\theta \in \mathbb{R}^d \mid \sum_{s \in \mathcal{S}} \pi_\theta(s) = 1\}$, we have $D_{\psi_k}(\theta, \theta_k) = \mathrm{KL}(\pi_\theta(s), \pi_{\theta_k}(s)) = \sum_{s \in \mathcal{S}} \pi_\theta(s) \log(\frac{\pi_\theta(s)}{\pi_{\theta_k}(s)})$, which is the KL divergence. Then our BGPO will become a momentum version of mirror descent policy optimization (Tomar et al., 2020).

### 4.2    VR-BGPO ALGORITHM

In the subsection, we propose a faster variance-reduced Bregman gradient policy optimization (VR-BGPO) algorithm based on a variance-reduced technique. The pseudo code of VR-BGPO algorithm is provided in Algorithm 2.

Consider the problem (4) is non-oblivious that the distribution $p(\tau|\theta)$ depends on the variable $\theta$ varying through the whole optimization procedure, we apply the importance sampling weight (Papini et al., 2018; Xu et al., 2019a) in estimating our policy gradient $u_{k+1}$, defined as

$$w(\tau_{k+1}|\theta_k, \theta_{k+1}) = \frac{p(\tau_{k+1}|\theta_k)}{p(\tau_{k+1}|\theta_{k+1})} = \prod_{t=0}^{H-1} \frac{\pi_{\theta_k}(a_t|s_t)}{\pi_{\theta_{k+1}}(a_t|s_t)}.$$

Except for different stochastic policy gradients $\{u_k\}$ and tuning parameters $\{\eta_k, \beta_k\}$ using in Algorithms 1 and 2, the steps 4 and 5 in these algorithms for updating parameter $\theta$ are the same. Interestingly, when choosing mirror mapping $\psi_k(\theta) = \frac{1}{2}\|\theta\|^2$, our VR-BGPO algorithm will reduce to a non-adaptive version of IS-MBPG algorithm (Huang et al., 2020).

## 5    CONVERGENCE ANALYSIS

In this section, we will analyze the convergence properties of our BGPO and VR-BGPO algorithms. All related proofs are provided in the Appendix A. Here we use the standard convergence metric $\|\mathcal{B}_{\lambda,\langle\theta,\nabla f(\theta_k)\rangle}^{\psi_k}(\theta)\|$ used in (Zhang & He, 2018; Yang et al., 2019) to evaluate the convergence Bregman gradient-based (a.k.a., mirror descent) algorithms. To give the convergence analysis, we first give some standard assumptions.

**Assumption 1.** *For function* $\log \pi_\theta(a|s)$, *its gradient and Hessian matrix are bounded,* i.e., *there exist constants* $C_g, C_h > 0$ *such that* $\|\nabla_\theta \log \pi_\theta(a|s)\| \leq C_g$, $\|\nabla_\theta^2 \log \pi_\theta(a|s)\| \leq C_h$.

**Assumption 2.** *Variance of stochastic gradient* $g(\tau|\theta)$ *is bounded,* i.e., *there exists a constant* $\sigma > 0$, *for all* $\pi_\theta$ *such that* $\mathbb{V}(g(\tau|\theta)) = \mathbb{E}\|g(\tau|\theta) - \nabla J(\theta)\|^2 \leq \sigma^2$.

**Assumption 3.** *For importance sampling weight* $w(\tau|\theta_1, \theta_2) = p(\tau|\theta_1)/p(\tau|\theta_2)$, *its variance is bounded,* i.e., *there exists a constant* $W > 0$, *it follows* $\mathbb{V}(w(\tau|\theta_1, \theta_2)) \leq W$ *for any* $\theta_1, \theta_2 \in \mathbb{R}^d$ *and* $\tau \sim p(\tau|\theta_2)$.

**Assumption 4.** *The function $J(\theta)$ has an upper bound in $\Theta$, i.e., $J^* = \sup_{\theta \in \Theta} J(\theta) < +\infty$.*

Assumptions 1 and 2 are commonly used in the PG algorithms (Papini et al., 2018; Xu et al., 2019a;b). Assumption 3 is widely used in the study of variance reduced PG algorithms (Papini et al., 2018; Xu et al., 2019a). In fact, the bounded importance sampling weight might be violated in some cases such as using neural networks as the policy. Thus, we can clip this importance sampling weights to guarantee the effectiveness of our algorithms as in (Papini et al., 2018). At the same time, the importance weights actually also have some nice properties, e.g., in soft-max policy it is bounded by $e^{c\|\theta_1 - \theta_2\|^2}$ for all $\theta_1, \theta_2 \in \Theta$. More recently, (Zhang et al., 2021) used a simple truncated update to relieve this uncheckable importance weight assumption. Assumption 4 guarantees the feasibility of the problem (4). Note that Assumptions 2 and 4 are satisfied automatically given Assumption 1 and the fact that all the rewards are bounded, i.e., $|r(s, a)| \leq R$ for any $s \in \mathcal{S}$ and $a \in \mathcal{A}$. For example, due to $|r(s, a)| \leq R$, we have $|J(\theta)| \leq \frac{R}{1-\gamma}$. So we have $J^* = \frac{R}{1-\gamma}$.

## 5.1 CONVERGENCE ANALYSIS OF BGPO ALGORITHM

In the subsection, we provide convergence properties of the BGPO algorithm. The detailed proof is provided in Appendix A.1.

**Theorem 1.** *Assume the sequence $\{\theta_k\}_{k=1}^K$ be generated from Algorithm 1. Let $\eta_k = \frac{b}{(m+k)^{1/2}}$ for all $k \geq 1$, $0 < \lambda \leq \frac{\nu m^{1/2}}{9Lb}$, $b > 0$, $\frac{8L\lambda}{\nu} \leq c \leq \frac{m^{1/2}}{b}$, and $m \geq \max\{b^2, (cb)^2\}$, we have*

$$\frac{1}{K}\sum_{k=1}^K \mathbb{E}\|\mathcal{B}_{\lambda, \langle \nabla f(\theta_k), \theta \rangle}^{\psi_k}(\theta_k)\| \leq \frac{2\sqrt{2M}m^{1/4}}{K^{1/2}} + \frac{2\sqrt{2M}}{K^{1/4}},$$

*where $M = \frac{J^* - J(\theta_1)}{\nu\lambda b} + \frac{\sigma^2}{\nu\lambda Lb} + \frac{m\sigma^2}{\nu\lambda Lb}\ln(m+K)$.*

**Remark 1.** *Without loss of generality, let $b = O(1)$, $m = O(1)$ and $\lambda = O(1)$, we have $M = O(\ln(m+K)) = \tilde{O}(1)$. Theorem 1 shows that the BGPO algorithm has a convergence rate of $\tilde{O}(\frac{1}{K^{1/4}})$. Let $K^{-\frac{1}{4}} \leq \epsilon$, we have $K = \tilde{O}(\epsilon^{-4})$. Since the BGPO algorithm only needs one trajectory to estimate the stochastic policy gradient at each iteration and runs $K$ iterations, it has the sample complexity of $1 \cdot K = O(\epsilon^{-4})$ for finding an $\epsilon$-stationary point.*

## 5.2 CONVERGENCE ANALYSIS OF VR-BGPO ALGORITHM

In the subsection, we give convergence properties of the VR-BGPO algorithm. The detailed proof is provided in Appendix A.2.

**Theorem 2.** *Suppose the sequence $\{\theta_k\}_{k=1}^K$ be generated from Algorithm 2. Let $\eta_k = \frac{b}{(m+k)^{1/3}}$ for all $k \geq 0$, $0 < \lambda \leq \frac{\nu m^{1/3}}{5\hat{L}b}$, $b > 0$, $c \in \left[\frac{2}{3b^3} + \frac{20\hat{L}^2\lambda^2}{\nu^2}, \frac{m^{2/3}}{b^2}\right]$ and $m \geq \max\left(2, b^3, (cb)^3, \left(\frac{5}{6b}\right)^{2/3}\right)$, we have*

$$\frac{1}{K}\sum_{k=1}^K \mathbb{E}\|\mathcal{B}_{\lambda, \langle \nabla f(\theta_k), \theta \rangle}^{\psi_k}(\theta_k)\| \leq \frac{2\sqrt{2M'}m^{1/6}}{K^{1/2}} + \frac{2\sqrt{2M'}}{K^{1/3}}, \tag{15}$$

*where $M' = \frac{J^* - J(\theta_1)}{b\nu\lambda} + \frac{m^{1/3}\sigma^2}{16b^2\hat{L}^2\lambda^2} + \frac{c^2\sigma^2 b^2}{8\hat{L}^2\lambda^2}$, $\hat{L}^2 = L^2 + 2G^2 C_w^2$, $G = C_g R/(1-\gamma)^2$ and $C_w = \sqrt{H(2HC_g^2 + C_h)(W+1)}$.*

**Remark 2.** *Without loss of generality, let $b = O(1)$, $m = O(1)$ and $\lambda = O(1)$, we have $M = O(\ln(m+K)) = \tilde{O}(1)$. Theorem 2 shows that the VR-BGPO algorithm has a convergence rate of $\tilde{O}(\frac{1}{K^{1/3}})$. Let $K^{-\frac{1}{3}} \leq \epsilon$, we have $K = (\epsilon^{-3})$. Since the VR-BGPO algorithm only needs one trajectory to estimate the stochastic policy gradient at each iteration and runs $K$ iterations, it reaches a lower sample complexity of $1 \cdot K = \tilde{O}(\epsilon^{-3})$ for finding an $\epsilon$-stationary point.*

## 6 EXPERIMENTS

In this section, we conduct some RL tasks to verify the effectiveness of our methods. We first study the effect of different choices of Bregman divergences with our algorithms (BGPO and VR-BGPO), and then we compare our VR-BGPO algorithm with other state-of-the-art methods such as TRPO (Schulman et al., 2015), PPO (Schulman et al., 2017), ProxHSPGA (Pham et al., 2020), VRMPO (Yang et al., 2019), and MDPO (Tomar et al., 2020). Our code is available at https://github.com/gaosh/BGPO.

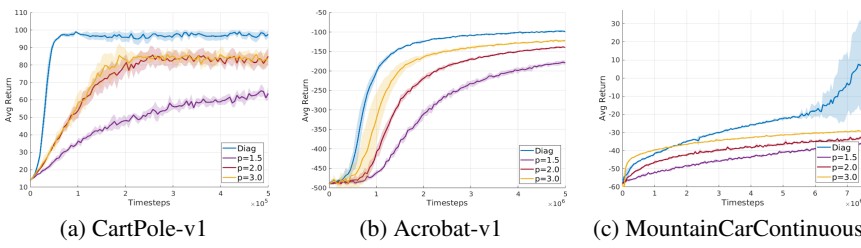

(a) CartPole-v1          (b) Acrobat-v1          (c) MountainCarContinuous

Figure 1: Effects of two Bregman Divergences: $l_p$-norm and diagonal term (Diag).

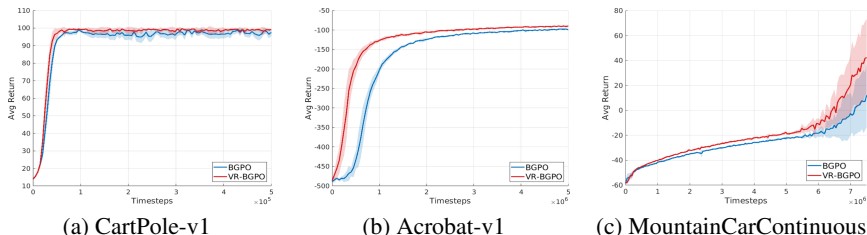

(a) CartPole-v1          (b) Acrobat-v1          (c) MountainCarContinuous

Figure 2: Comparison between BGPO and VR-BGPO on different environments.

## 6.1 EFFECTS OF BREGMAN DIVERGENCES

In the subsection, we examine how different Bregman divergences affect the performance of our algorithms. In the first setting, we let mirror mapping $\psi_k(x) = \|x\|_p$ $(p \geq 1)$ with different $p$ to test the performance our algorithms. Let $\psi_k^*(y) = (\sum_{i=1}^d |y_i|^q)^{\frac{1}{q}}$ be the conjugate mapping of $\psi_k(x)$, where $p^{-1} + q^{-1} = 1$, $p, q > 1$. According to (Beck & Teboulle, 2003), when $\Theta = \mathbb{R}^d$, the update of $\tilde{\theta}_{k+1}$ in our algorithms can be calculated by $\tilde{\theta}_{k+1} = \nabla \psi_k^*(\nabla \psi_k(\theta_k) + \lambda u_k)$, where $\nabla \psi_k(x_j)$ and $\nabla \psi_k^*(y_j)$ are $p$-norm link functions, and $\nabla \psi_k(x_j) = \frac{\text{sign}(x_j)|x_j|^{p-1}}{\|x\|_p^{p-2}}$, $\nabla \psi_k^*(y_j) = \frac{\text{sign}(y_j)|y_j|^{q-1}}{\|y\|_q^{q-2}}$, and $j$ is the coordinate index of $x$ and $y$. In the second setting, we apply diagonal term on the mirror mapping $\psi_k(x) = \frac{1}{2}x^T M_k x$, where $M_k$ is a diagonal matrix with positive values. In the experiments, we generate $H_k = \text{diag}(\sqrt{v_k} + \alpha)$, $v_k = \beta v_{k-1} + (1-\beta)u_k^2$, and $\alpha > 0, \beta \in (0, 1)$, as in Super-Adam algorithm (Kingma & Ba, 2014; Huang et al., 2021). Then we have $D_{\psi_k}(y, x) = \frac{1}{2}(y-x)^T H_k(y-x)$. Under this setting, the update of $\tilde{\theta}_{k+1}$ can also be analytically solved $\tilde{\theta}_{k+1} = \theta_k - \lambda H_k^{-1} u_k$.

To test the effectiveness of two different Bregman divergences, we evaluate them on three classic control environments from gym Brockman et al. (2016): CartPole-v1, Acrobat-v1, and MountainCarContinuous-v0. In the experiment, categorical policy is used for CartPole and Acrobot environments, and Gaussian policy is used for MountainCar. Gaussian value functions are used in all settings. All policies and value functions are parameterized by multilayer perceptrons (MLPs). For a fair comparison, all settings use the same initialization for policies. We run each setting five times and plot the mean and variance of average returns. For $l_p$-norm mapping, we test three different values of $p = (1.50, 2.0, 3.0)$. For diagonal mapping, we set $\beta = 0.999$ and $\alpha = 10^{-8}$. We set hyperparameters $\{b, m, c\}$ to be the same. $\lambda$ still needs to be tuned for different $p$ to achieve relatively good performance. For simplicity, we use BGPO-Diag to represent BGPO with diagonal mapping, and we use BGPO-$l_p$ to represent BGPO with $l_p$-norm mapping. Details about the setup of environments and hyperparameters are provided in the Appendix C.

From Fig. 1, we can find that BGPO-Diag largely outperforms BGPO-$l_p$ with different choices of $p$. The parameter tuning of BGPO-$l_p$ is much more difficult than BGPO-Diag because each $p$ requires an individual $\lambda$ to achieve the desired performance.

## 6.2 COMPARISON BETWEEN BGPO AND VR-BGPO

To understand the effectiveness of variance reduced technique used in our VR-BGPO algorithm, we compare BGPO and VR-BGPO using the same settings introduced in section. 6.1. Both algorithms use the diagonal mapping for $\psi$, since it performs much better than $l_p$-norm. From Fig. 2 given in the Appendix C, we can see that VR-BGPO can outperform BGPO in all three environments.

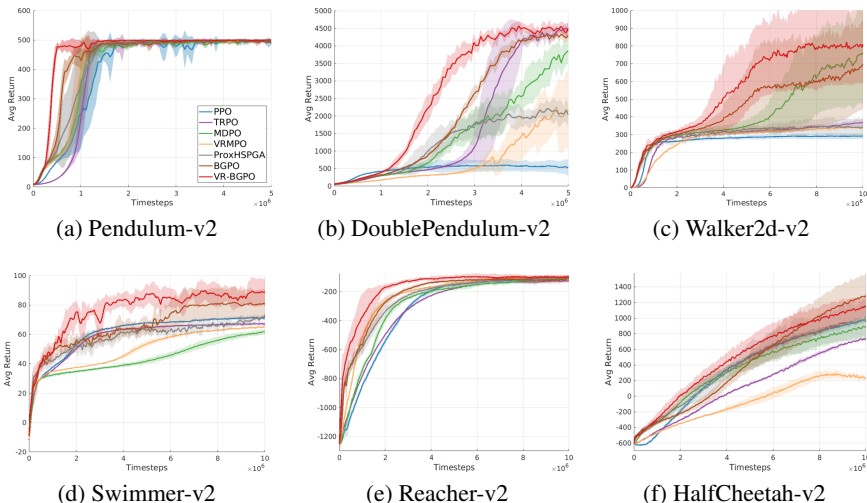

Figure 3: Experimental results of our algorithms and other baseline algorithms on six environments.

In CartPole, both algorithms converge very fast and have similar performance, and VR-BGPO is more stable than BGPO. The advantage of VR-BGPO becomes large in Acrobot and MountainCar environments, probably because the task is more difficult compared to CartPole.

### 6.3    COMPARE TO OTHER METHODS

In this subsection, we apply our BGPO and VR-BGPO algorithms to compare with the other methods. For our BGPO and VR-BGPO, we use diagonal mapping for $\psi$. For VRMPO, we follow their implementation and use $l_p$-norm for $\psi$. For MDPO, $\psi$ is the negative Shannon entropy, and the Bregman divergence becomes KL-divergence.

To evaluate the performance of these algorithms, we test them on six gym (Brockman et al., 2016) environments with continuous control tasks: Inverted-DoublePendulum-v2, Walker2d-v2, Reacher-v2, Swimmer-v2, Inverted-Pendulum-v2 and HalfCheetah-v2. We use Gaussian policies and Gaussian value functions for all environments, and both of them are parameterized by MLPs. To ensure a fair comparison, all policies use the same initialization. For TRPO and PPO, we use the implementations provided by garage (garage contributors, 2019). We carefully implement MDPO and VRMPO following the description provided by the original papers. All methods include our method, are implemented with garage (garage contributors, 2019) and pytorch (Paszke et al., 2019). We run all algorithms ten times on each environment and report the mean and variance of average returns. Details about the setup of environments and hyperparameters are also provided in the Appendix C.

From Fig. 3, we can find that our VR-BGPO method consistently outperforms all the other methods. Our BGPO basically reaches the second best performances. From the results of our BGPO, we can find that using a proper Bregman (mirror) distance can improve performances of the PG methods. From the results of our VR-BGPO, we can find that using a proper variance-reduced technique can further improve performances of the BGPO. ProxHSPGA can reach some relatively good performances by using the variance reduced technique. MDPO can achieve good results in some environments, but it can not outperform PPO or TRPO in Swimmer and InvertedDoublePendulum. VRMPO only outperforms PPO and TRPO in Reacher and InvertedDoublePendulum. The undesirable performance of VRMPO is probably because it uses $l_p$ norm for $\psi$, which requires careful tuning of learning rate.

## 7    CONCLUSION

In the paper, we proposed a novel Bregman gradient policy optimization framework for reinforcement learning based on Bregman divergences and momentum techniques. Moreover, we studied convergence properties of the proposed methods under the nonconvex setting.

### ACKNOWLEDGMENT

This work was partially supported by NSF IIS 1845666, 1852606, 1838627, 1837956, 1956002, OIA 2040588.

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

## A APPENDIX

In this section, we study the convergence properties of our algorithms. We first provide some useful lemmas.

**Lemma 1.** *(Proposition 4.2 in Xu et al. (2019b)) Suppose $g(\tau|\theta)$ is the PGT estimator. Under Assumption 1, we have*

1) *$g(\tau|\theta)$ is L-Lipschitz differential, i.e., $\|g(\tau|\theta_1) - g(\tau|\theta_2)\| \leq L\|\theta_1 - \theta_2\|$ for all $\theta_1, \theta_2 \in \Theta$, where $L = C_h R/(1-\gamma)^2$;*

2) *$J(\theta)$ is L-smooth, i.e., $\|\nabla^2 J(\theta)\| \leq L$;*

3) *$g(\tau|\theta)$ is bounded, i.e., $\|g(\tau|\theta)\| \leq G$ for all $\theta \in \Theta$ with $G = C_g R/(1-\gamma)^2$.*

**Lemma 2.** *(Lemma 6.1 in Xu et al. (2019a)) Under Assumptions 1 and 3, let $w(\tau|\theta_{k-1}, \theta_k) = g(\tau|\theta_{k-1})/g(\tau|\theta_k)$, we have*

$$\mathbb{V}[w(\tau|\theta_{k-1}, \theta_k)] \leq C_w^2 \|\theta_k - \theta_{k-1}\|^2, \tag{16}$$

*where $C_w = \sqrt{H(2HC_g^2 + C_h)(W+1)}$.*

**Lemma 3.** *(Lemma 1 in (Ghadimi et al., 2016)) Let $\mathcal{X} \subseteq \mathbb{R}^d$ be a closed convex set, and $\phi : \mathcal{X} \to \mathbb{R}$ be a convex function but possibly nonsmooth, and $D_\psi : \mathcal{X} \times \mathcal{X} \to \mathbb{R}$ is Bregman divergence related to the $\nu$-strongly convex function $\psi$. Then we define*

$$x^+ = \arg\min_{z \in \mathcal{X}} \left\{ \langle g, z \rangle + \frac{1}{\lambda} D_\psi(z, x) + \phi(z) \right\}, \ \forall x \in \mathcal{X} \tag{17}$$

$$P_\mathcal{X}(x, g, \lambda) = \frac{1}{\lambda}(x - x^+), \tag{18}$$

*where $g \in \mathbb{R}^d$, $\lambda > 0$ and $D_\psi(z, x) = \psi(z) - \big(\psi(x) + \langle \nabla\psi(x), z - x \rangle\big)$. Then the following statement holds*

$$\langle g, P_\mathcal{X}(x, g, \lambda) \rangle \geq \nu\|P_\mathcal{X}(x, g, \lambda)\|^2 + \frac{1}{\lambda}\big[\phi(x^+) - \phi(x)\big]. \tag{19}$$

**Lemma 4.** *(Proposition 1 in Ghadimi et al. (2016)) Let $x_1^+$ and $x_2^+$ be given in (17) with g replaced by $g_1$ and $g_2$ respectively. Then let $P_\mathcal{X}(x, g_1, \lambda)$ and $P_\mathcal{X}(x, g_2, \lambda)$ be defined in (18) with $x^+$ replaced by $x_1^+$ and $x_2^+$ respectively. we have*

$$\|P_\mathcal{X}(x, g_1, \lambda) - P_\mathcal{X}(x, g_2, \lambda)\| \leq \frac{1}{\nu}\|g_1 - g_2\|. \tag{20}$$

**Lemma 5.** *(Lemma 1 in (Cortes et al., 2010)) Let $w(x) = \frac{P(x)}{Q(x)}$ be the importance weight for distributions $P$ and $Q$. The following identities hold for the expectation, second moment, and variance of $w(x)$*

$$\mathbb{E}[w(x)] = 1, \ \mathbb{E}[w^2(x)] = d_2(P\|Q),$$
$$\mathbb{V}[w(x)] = d_2(P\|Q) - 1, \tag{21}$$

*where $d_2(P\|Q) = 2^{D(P\|Q)}$, and $D(P\|Q)$ is Rényi divergence between distributions $P$ and $Q$.*

**Lemma 6.** *Suppose that the sequence $\{\theta_k\}_{k=1}^K$ be generated from Algorithms 1 or 2. Let $0 < \eta_k \leq 1$ and $0 < \lambda \leq \frac{\nu}{2L\eta_k}$, then we have*

$$f(\theta_{k+1}) - f(\theta_k) \leq \frac{\eta_k \lambda}{\nu}\|\nabla f(\theta_k) - u_k\|^2 - \frac{\nu\eta_k}{2\lambda}\|\tilde{\theta}_{k+1} - \theta_k\|^2. \tag{22}$$

*Proof.* According to Assumption 1 and Lemma 1, the function $f(\theta)$ is L-smooth. Then we have

$$f(\theta_{k+1}) \leq f(\theta_k) + \langle \nabla f(\theta_k), \theta_{k+1} - \theta_k \rangle + \frac{L}{2}\|\theta_{k+1} - \theta_k\|^2 \tag{23}$$

$$= f(\theta_k) + \eta_k \langle \nabla f(\theta_k), \tilde{\theta}_{k+1} - \theta_k \rangle + \frac{L\eta_k^2}{2}\|\tilde{\theta}_{k+1} - \theta_k\|^2$$

$$= f(\theta_k) + \eta_k \langle \nabla f(\theta_k) - u_k, \tilde{\theta}_{k+1} - \theta_k \rangle + \eta_k \langle u_k, \tilde{\theta}_{k+1} - \theta_k \rangle + \frac{L\eta_k^2}{2}\|\tilde{\theta}_{k+1} - \theta_k\|^2,$$

where the second equality is due to $\theta_{k+1} = \theta_k + \eta_k(\tilde{\theta}_{k+1} - \theta_k)$. By the step 4 of Algorithm 1 or 2, we have $\tilde{\theta}_{k+1} = \arg\min_{\theta \in \Theta} \left\{ \langle u_k, \theta \rangle + \frac{1}{\lambda} D_{\psi_k}(\theta, \theta_k) \right\}$. By using Lemma 3 with $\phi(\cdot) = 0$, we have

$$\langle u_k, \frac{1}{\lambda}(\theta_k - \tilde{\theta}_{k+1}) \rangle \geq \nu \| \frac{1}{\lambda}(\theta_k - \tilde{\theta}_{k+1}) \|^2. \tag{24}$$

Thus, we can obtain

$$\langle u_k, \tilde{\theta}_{k+1} - \theta_k \rangle \leq -\frac{\nu}{\lambda} \| \tilde{\theta}_{k+1} - \theta_k \|^2. \tag{25}$$

According to the Cauchy-Schwarz inequality and Young's inequality, we have

$$\langle \nabla f(\theta_k) - u_k, \tilde{\theta}_{k+1} - \theta_k \rangle \leq \| \nabla f(\theta_k) - u_k \| \| \tilde{\theta}_{k+1} - \theta_k \|$$
$$\leq \frac{\lambda}{\nu} \| \nabla f(\theta_k) - u_k \|^2 + \frac{\nu}{4\lambda} \| \tilde{\theta}_{k+1} - \theta_k \|^2. \tag{26}$$

Combining the inequalities (23), (25) with (26), we obtain

$$f(\theta_{k+1}) \leq f(\theta_k) + \eta_k \langle \nabla f(\theta_k) - u_k, \tilde{\theta}_{k+1} - \theta_k \rangle + \eta_k \langle u_k, \tilde{\theta}_{k+1} - \theta_k \rangle + \frac{L\eta_k^2}{2} \| \tilde{\theta}_{k+1} - \theta_k \|^2$$

$$\leq f(\theta_k) + \frac{\eta_k \lambda}{\nu} \| \nabla f(\theta_k) - u_k \|^2 + \frac{\nu \eta_k}{4\lambda} \| \tilde{\theta}_{k+1} - \theta_k \|^2 - \frac{\nu \eta_k}{\lambda} \| \tilde{\theta}_{k+1} - \theta_k \|^2 + \frac{L\eta_k^2}{2} \| \tilde{\theta}_{k+1} - \theta_k \|^2$$

$$= f(\theta_k) + \frac{\eta_k \lambda}{\nu} \| \nabla f(\theta_k) - u_k \|^2 - \frac{\nu \eta_k}{2\lambda} \| \tilde{\theta}_{k+1} - \theta_k \|^2 - \left( \frac{\nu \eta_k}{4\lambda} - \frac{L\eta_k^2}{2} \right) \| \tilde{\theta}_{k+1} - \theta_k \|^2$$

$$\leq f(\theta_k) + \frac{\eta_k \lambda}{\nu} \| \nabla f(\theta_k) - u_k \|^2 - \frac{\nu \eta_k}{2\lambda} \| \tilde{\theta}_{k+1} - \theta_k \|^2, \tag{27}$$

where the last inequality is due to $0 < \lambda \leq \frac{\nu}{2L\eta_k}$. $\qquad \square$

## A.1 CONVERGENCE ANALYSIS OF BGPO ALGORITHM

In this subsection, we analyze the convergence properties of BGPO algorithm.

**Lemma 7.** *Assume the stochastic policy gradient $u_{k+1}$ be generated from Algorithm 1, given $0 < \beta_k \leq 1$, we have*

$$\mathbb{E}\| \nabla f(\theta_{k+1}) - u_{k+1} \|^2 \leq (1 - \beta_{k+1}) \mathbb{E}\| \nabla f(\theta_k) - u_k \|^2 + \frac{2}{\beta_{k+1}} L^2 \eta_k^2 \mathbb{E}\| \tilde{\theta}_{k+1} - \theta_k \|^2 + \beta_{k+1}^2 \sigma^2.$$

*Proof.* By the definition of $u_{k+1}$ in Algorithm 1, we have

$$u_{k+1} - u_k = -\beta_{k+1} u_k - \beta_{k+1} g(\tau_{k+1}|\theta_{k+1}). \tag{28}$$

Since $\nabla f(\theta_k) = -J(\theta_k)$ for all $k \geq 1$, we have

$\mathbb{E}\| \nabla f(\theta_{k+1}) - u_{k+1} \|^2$

$= \mathbb{E}\| -\nabla J(\theta_k) - u_k - \nabla J(\theta_{k+1}) + \nabla J(\theta_k) - (u_{k+1} - u_k) \|^2$

$= \mathbb{E}\| -\nabla J(\theta_k) - u_k - \nabla J(\theta_{k+1}) + \nabla J(\theta_k) + \beta_{k+1} u_k + \beta_{k+1} g(\tau_{k+1}|\theta_{k+1}) \|^2$

$= \mathbb{E}\| (1 - \beta_{k+1})(-\nabla J(\theta_k) - u_k) + \beta_{k+1}(-\nabla J(\theta_{k+1}) + g(\tau_{k+1}|\theta_{k+1}))$
$\quad + (1 - \beta_{k+1})\big( -\nabla J(\theta_{k+1}) + \nabla J(\theta_k) \big) \|^2$

$= (1 - \beta_{k+1})^2 \mathbb{E}\| \nabla J(\theta_k) + u_k + \nabla J(\theta_{k+1}) - \nabla J(\theta_k) \|^2 + \beta_{k+1}^2 \mathbb{E}\| \nabla J(\theta_{k+1}) - g(\tau_{k+1}|\theta_{k+1}) \|^2$

$\leq (1 - \beta_{k+1})^2 (1 + \beta_{k+1}) \mathbb{E}\| \nabla J(\theta_k) + u_k \|^2 + (1 - \beta_{k+1})^2 (1 + \frac{1}{\beta_{k+1}}) \mathbb{E}\| \nabla J(\theta_{k+1}) - \nabla J(\theta_k) \|^2$
$\quad + \beta_{k+1}^2 \mathbb{E}\| \nabla J(\theta_{k+1}) - g(\tau_{k+1}|\theta_{k+1}) \|^2$

$\leq (1 - \beta_{k+1}) \mathbb{E}\| \nabla J(\theta_k) + u_k \|^2 + \frac{2}{\beta_{k+1}} \mathbb{E}\| \nabla J(\theta_{k+1}) - \nabla J(\theta_k) \|^2 + \beta_{k+1}^2 \mathbb{E}\| \nabla J(\theta_{k+1}) - g(\tau_{k+1}|\theta_{k+1}) \|^2$

$\leq (1 - \beta_{k+1}) \mathbb{E}\| \nabla J(\theta_k) + u_k \|^2 + \frac{2}{\beta_{k+1}} L^2 \mathbb{E}\| \theta_{k+1} - \theta_k \|^2 + \beta_{k+1}^2 \mathbb{E}\| \nabla J(\theta_{k+1}) - g(\tau_{k+1}|\theta_{k+1}) \|^2$

$\leq (1 - \beta_{k+1}) \mathbb{E}\| \nabla f(\theta_k) - u_k \|^2 + \frac{2}{\beta_{k+1}} L^2 \eta_k^2 \mathbb{E}\| \tilde{\theta}_{k+1} - \theta_k \|^2 + \beta_{k+1}^2 \sigma^2, \tag{29}$

where the fourth equality holds by $\mathbb{E}_{\tau_{k+1} \sim p(\tau | \theta_{k+1})}[g(\tau_{k+1} | \theta_{k+1})] = \nabla J(\theta_{k+1})$; the first inequality holds by Young's inequality; the second inequality is due to $0 < \beta_{k+1} \leq 1$ such that $(1 - \beta_{k+1})^2 (1 + \beta_{k+1}) = 1 - \beta_{k+1} - \beta_{k+1}^2 + \beta_{k+1}^3 \leq 1 - \beta_{k+1}$ and $(1 - \beta_{k+1})^2 (1 + \frac{1}{\beta_{k+1}}) \leq 1 + \frac{1}{\beta_{k+1}} \leq \frac{2}{\beta_{k+1}}$; the last inequality holds by Assumption 2. □

**Theorem 3.** *Assume the sequence $\{\theta_k\}_{k=1}^K$ be generated from Algorithm 1. Let $\eta_k = \frac{b}{(m+k)^{1/2}}$ for all $k \geq 1$, $0 < \lambda \leq \frac{\nu m^{1/2}}{9Lb}$, $b > 0$, $\frac{8L\lambda}{\nu} \leq c \leq \frac{m^{1/2}}{b}$, and $m \geq \max\{b^2, (cb)^2\}$, we have*

$$\frac{1}{K} \sum_{k=1}^K \mathbb{E}\|\mathcal{B}_{\lambda, \langle \nabla f(\theta_k), \theta \rangle}^{\psi_k}(\theta_k)\| \leq \frac{2\sqrt{2M} m^{1/4}}{K^{1/2}} + \frac{2\sqrt{2M}}{K^{1/4}},$$

*where $M = \frac{J^* - J(\theta_1)}{\nu \lambda b} + \frac{\sigma^2}{\nu \lambda Lb} + \frac{m \sigma^2}{\nu \lambda Lb} \ln(m + K)$.*

*Proof.* Since $\eta_k = \frac{b}{(m+k)^{1/2}}$ is decreasing on $k$, we have $\eta_k \leq \eta_0 = \frac{b}{m^{1/2}} \leq 1$ for all $k \geq 0$. At the same time, let $m \geq (cb)^2$, we have $\beta_{k+1} = c\eta_k \leq c\eta_0 = \frac{cb}{m^{1/2}} \leq 1$. Consider $m \geq (cb)^2$, we have $c \leq \frac{m^{1/2}}{b}$. Since $0 < \lambda \leq \frac{\nu m^{1/2}}{9Lb}$, we have $\lambda \leq \frac{\nu m^{1/2}}{9Lb} \leq \frac{\nu m^{1/2}}{2Lb} = \frac{\nu}{2L\eta_0} \leq \frac{\nu}{2L\eta_k}$ for all $k \geq 0$. According to Lemma 7, we have

$$\mathbb{E}\|\nabla f(\theta_{k+1}) - u_{k+1}\|^2 - \mathbb{E}\|\nabla f(\theta_k) - u_k\|^2$$
$$\leq -\beta_{k+1} \mathbb{E}\|\nabla f(\theta_k) - u_k\|^2 + \frac{2}{\beta_{k+1}} L^2 \eta_k^2 \mathbb{E}\|\tilde{\theta}_{k+1} - \theta_k\|^2 + \beta_{k+1}^2 \sigma^2$$
$$= -c\eta_k \mathbb{E}\|\nabla f(\theta_k) - u_k\|^2 + \frac{2L^2}{c} \eta_k \mathbb{E}\|\tilde{\theta}_{k+1} - \theta_k\|^2 + c^2 \eta_k^2 \sigma^2$$
$$= -\frac{8L\lambda}{\nu} \eta_k \mathbb{E}\|\nabla f(\theta_k) - u_k\|^2 + \frac{L\nu}{4\lambda} \eta_k \mathbb{E}\|\tilde{\theta}_{k+1} - \theta_k\|^2 + \frac{m \eta_k^2 \sigma^2}{b^2}, \tag{30}$$

where the first equality is due to $\beta_{k+1} = c\eta_k$ and the last equality holds by $\frac{8L\lambda}{\nu} \leq c \leq \frac{m^{1/2}}{b}$.

Next we define a *Lyapunov* function $\Phi_k = \mathbb{E}\left[f(\theta_k) + \frac{1}{L}\|\nabla f(\theta_k) - u_k\|^2\right]$ for any $t \geq 1$. Then we have

$$\Phi_{k+1} - \Phi_k = \mathbb{E}\left[f(\theta_{k+1}) - f(\theta_k) + \frac{1}{L}\left(\|\nabla f(\theta_{k+1}) - u_{k+1}\|^2 - \|\nabla f(\theta_k) - u_k\|^2\right)\right]$$
$$\leq \frac{\lambda \eta_k}{\nu} \mathbb{E}\|\nabla f(\theta_k) - u_k\|^2 - \frac{\nu \eta_k}{2\lambda} \mathbb{E}\|\tilde{\theta}_{k+1} - \theta_k\|^2 - \frac{8\lambda \eta_k}{\nu} \mathbb{E}\|\nabla f(\theta_k) - u_k\|^2$$
$$+ \frac{\nu \eta_k}{4\lambda} \mathbb{E}\|\tilde{\theta}_{k+1} - \theta_k\|^2 + \frac{m \eta_k^2 \sigma^2}{Lb^2}$$
$$\leq -\frac{\lambda}{4\nu} \eta_k \mathbb{E}\|\nabla f(\theta_k) - u_k\|^2 - \frac{\nu}{4\lambda} \eta_k \mathbb{E}\|\tilde{\theta}_{k+1} - \theta_k\|^2 + \frac{m \eta_k^2 \sigma^2}{Lb^2}, \tag{31}$$

where the first inequality follows by the Lemma 6 and the above inequality (30).

Summing the above inequality (31) over $k$ from 1 to $K$, we can obtain

$$\sum_{k=1}^{K} \mathbb{E}[\frac{\lambda}{4\nu}\eta_k \mathbb{E}\|\nabla f(\theta_k) - u_k\|^2 + \frac{\nu}{4\lambda}\eta_k \mathbb{E}\|\tilde{\theta}_{k+1} - \theta_k\|^2]$$

$$\leq \Phi_1 - \Phi_{K+1} + \frac{m\sigma^2}{Lb^2}\sum_{k=1}^{K}\eta_k^2$$

$$= f(\theta_1) - f(\theta_{K+1}) + \frac{1}{L}\|\nabla f(\theta_1) - u_1\|^2 - \frac{1}{L}\|\nabla f(\theta_{K+1}) - u_{K+1}\|^2 + \frac{m\sigma^2}{Lb^2}\sum_{k=1}^{K}\eta_k^2$$

$$\leq J(\theta_{K+1}) - J(\theta_1) + \frac{1}{L}\|\nabla J(\theta_1) - g(\tau_1|\theta_1)\|^2 + \frac{m\sigma^2}{Lb^2}\sum_{k=1}^{K}\eta_k^2$$

$$\leq J^* - J(\theta_1) + \frac{\sigma^2}{L} + \frac{m\sigma^2}{L}\int_1^K \frac{1}{m+k}dk$$

$$\leq J^* - J(\theta_1) + \frac{\sigma^2}{L} + \frac{m\sigma^2}{L}\ln(m+K), \tag{32}$$

where the last second inequality holds by Assumptions 2 and 4.

Since $\eta_k$ is decreasing, we have

$$\frac{1}{K}\sum_{k=1}^{K}\mathbb{E}[\frac{1}{4\nu^2}\mathbb{E}\|\nabla f(\theta_k) - u_k\|^2 + \frac{1}{4\lambda^2}\mathbb{E}\|\tilde{\theta}_{k+1} - \theta_k\|^2]$$

$$\leq \frac{J^* - J(\theta_1)}{K\nu\lambda\eta_K} + \frac{\sigma^2}{K\nu\lambda L\eta_K} + \frac{m\sigma^2}{\nu\lambda LK\eta_K}\ln(m+K)$$

$$\leq \left(\frac{J^* - J(\theta_1)}{\nu\lambda b} + \frac{\sigma^2}{\nu\lambda Lb} + \frac{m\sigma^2}{\nu\lambda Lb}\ln(m+K)\right)\frac{(m+K)^{1/2}}{K}. \tag{33}$$

Let $M = \frac{J^* - J(\theta_1)}{\nu\lambda b} + \frac{\sigma^2}{\nu\lambda Lb} + \frac{m\sigma^2}{\nu\lambda Lb}\ln(m+K)$, the above inequality (33) reduces to

$$\frac{1}{K}\sum_{k=1}^{K}\mathbb{E}[\frac{1}{4\nu^2}\mathbb{E}\|\nabla f(\theta_k) - u_k\|^2 + \frac{1}{4\lambda^2}\mathbb{E}\|\tilde{\theta}_{k+1} - \theta_k\|^2] \leq \frac{M}{K}(m+K)^{1/2}. \tag{34}$$

According to Jensen's inequality, we have

$$\frac{1}{K}\sum_{k=1}^{K}\mathbb{E}\big[\frac{1}{2\nu}\|\nabla f(\theta_k) - u_k\| + \frac{1}{2\lambda}\|\tilde{\theta}_{k+1} - \theta_k\|\big]$$

$$\leq \big(\frac{2}{K}\sum_{k=1}^{K}\mathbb{E}\big[\frac{1}{4\nu^2}\|\nabla f(\theta_k) - u_k\|^2 + \frac{1}{4\lambda^2}\|\tilde{\theta}_{k+1} - \theta_k\|^2\big]\big)^{1/2}$$

$$\leq \frac{\sqrt{2M}}{K^{1/2}}(m+K)^{1/4} \leq \frac{\sqrt{2M}m^{1/4}}{K^{1/2}} + \frac{\sqrt{2M}}{K^{1/4}}, \tag{35}$$

where the last inequality is due to the inequality $(a+b)^{1/4} \leq a^{1/4} + b^{1/4}$ for all $a, b \geq 0$. Thus we have

$$\frac{1}{K}\sum_{k=1}^{K}\mathbb{E}\big[\frac{1}{\nu}\|\nabla f(\theta_k) - u_k\| + \frac{1}{\lambda}\|\tilde{\theta}_{k+1} - \theta_k\|\big] \leq \frac{2\sqrt{2M}m^{1/4}}{K^{1/2}} + \frac{2\sqrt{2M}}{K^{1/4}}. \tag{36}$$

By the step 4 of Algorithm 1, we have

$$\mathcal{B}^{\psi_k}_{\lambda,\langle u_k,\theta\rangle}(\theta_k) = P_\Theta(\theta_k, u_k, \lambda) = \frac{1}{\lambda}\big(\theta_k - \tilde{\theta}_{k+1}\big). \tag{37}$$

At the same time, as in Ghadimi et al. (2016), we define

$$\mathcal{B}^{\psi_k}_{\lambda,\langle\nabla f(\theta_k),\theta\rangle}(\theta_k) = P_\Theta(\theta_k,\nabla f(\theta_k),\lambda) = \frac{1}{\lambda}\big(\theta_k - \theta^+_{k+1}\big), \tag{38}$$

where

$$\theta^+_{k+1} = \arg\min_{\theta\in\Theta}\big\{\langle\nabla f(\theta_k),\theta\rangle + \frac{1}{\lambda}D_{\psi_k}(\theta,\theta_k)\big\}. \tag{39}$$

According to the above Lemma 4, we have $\|\mathcal{B}^{\psi_k}_{\lambda,\langle u_k,\theta\rangle}(\theta_k)-\mathcal{B}^{\psi_k}_{\lambda,\langle\nabla f(\theta_k),\theta\rangle}(\theta_k)\| \le \frac{1}{\nu}\|u_k-\nabla f(\theta_k)\|$.
Then we have

$$\begin{aligned}
\|\mathcal{B}^{\psi_k}_{\lambda,\langle\nabla f(\theta_k),\theta\rangle}(\theta_k)\| &\le \|\mathcal{B}^{\psi_k}_{\lambda,\langle u_k,\theta\rangle}(\theta_k)\| + \|\mathcal{B}^{\psi_k}_{\lambda,\langle u_k,\theta\rangle}(\theta_k) - \mathcal{B}^{\psi_k}_{\lambda,\langle\nabla f(\theta_k),\theta\rangle}(\theta_k)\| \\
&\le \|\mathcal{B}^{\psi_k}_{\lambda,\langle u_k,\theta\rangle}(\theta_k)\| + \frac{1}{\nu}\|u_k - \nabla f(\theta_k)\| \\
&= \frac{1}{\lambda}\|\tilde{\theta}_{k+1} - \theta_k\| + \frac{1}{\nu}\|u_k - \nabla f(\theta_k)\|.
\end{aligned} \tag{40}$$

By the above inequalities (36) and (40), we have

$$\frac{1}{K}\sum_{k=1}^{K}\mathbb{E}\|\mathcal{B}^{\psi_k}_{\lambda,\langle\nabla f(\theta_k),\theta\rangle}(\theta_k)\| \le \frac{2\sqrt{2M}m^{1/4}}{K^{1/2}} + \frac{2\sqrt{2M}}{K^{1/4}}. \tag{41}$$

$\square$

## A.2 CONVERGENCE ANALYSIS OF VR-BGPO ALGORITHM

In this subsection, we will analyze convergence properties of the VR-BGPO algorithm.

**Lemma 8.** *Assume that the stochastic policy gradient $u_{k+1}$ be generated from Algorithm 2, given $0 < \beta_k \le 1$, we have*

$$\mathbb{E}\|\nabla f(\theta_{k+1}) - u_{k+1}\|^2 \le (1-\beta_{k+1})\mathbb{E}\|\nabla f(\theta_k) - u_k\|^2 + 4\hat{L}^2\eta_k^2\|\tilde{\theta}_{k+1} - \theta_k\|^2 + 2\beta_{k+1}^2\sigma^2,$$

*where $\hat{L}^2 = L^2 + 2G^2C_w^2$ and $C_w = \sqrt{H(2HC_g^2 + C_h)(W+1)}$.*

*Proof.* By the definition of $u_{k+1}$ in Algorithm 2, we have

$$u_{k+1} - u_k$$
$$= -\beta_{k+1}u_k - \beta_{k+1}g(\tau_{k+1}|\theta_{k+1}) + (1-\beta_{k+1})\big(-g(\tau_{k+1}|\theta_{k+1}) + w(\tau_{k+1}|\theta_k,\theta_{k+1})g(\tau_{k+1}|\theta_k)\big).$$

Since $\nabla f(\theta_{k+1}) = -\nabla J(\theta_{k+1})$, we have

$$\begin{aligned}
&\mathbb{E}\|\nabla f(\theta_{k+1}) - u_{k+1}\|^2 \\
&= \mathbb{E}\|\nabla J(\theta_k) + u_k + \nabla J(\theta_{k+1}) - \nabla J(\theta_k) + (u_{k+1} - u_k)\|^2 \\
&= \mathbb{E}\|\nabla J(\theta_k) + u_k + \nabla J(\theta_{k+1}) - \nabla J(\theta_k) - \beta_{k+1}u_k - \beta_{k+1}g(\tau_{k+1}|\theta_{k+1}) \\
&\quad + (1-\beta_{k+1})\big(-g(\tau_{k+1}|\theta_{k+1}) + w(\tau_{k+1}|\theta_k,\theta_{k+1})g(\tau_{k+1}|\theta_k)\big)\|^2 \\
&= \mathbb{E}\|(1-\beta_{k+1})(\nabla J(\theta_k) + u_k) + \beta_{k+1}(\nabla J(\theta_{k+1}) - g(\tau_{k+1}|\theta_{k+1})) \\
&\quad - (1-\beta_{k+1})\big(g(\tau_{k+1}|\theta_{k+1}) - w(\tau_{k+1}|\theta_k,\theta_{k+1})g(\tau_{k+1}|\theta_k) - (\nabla J(\theta_{k+1}) - \nabla J(\theta_k))\big)\|^2 \\
&= (1-\beta_{k+1})^2\mathbb{E}\|\nabla J(\theta_k) + u_k\|^2 + \mathbb{E}\|\beta_{k+1}(\nabla J(\theta_{k+1}) - g(\tau_{k+1}|\theta_{k+1})) \\
&\quad - (1-\beta_{k+1})\big(g(\tau_{k+1}|\theta_{k+1}) - w(\tau_{k+1}|\theta_k,\theta_{k+1})g(\tau_{k+1}|\theta_k) - (\nabla J(\theta_{k+1}) - \nabla J(\theta_k))\big)\|^2 \\
&\le (1-\beta_{k+1})^2\mathbb{E}\|\nabla J(\theta_k) + u_k\|^2 + 2\beta_{k+1}^2\mathbb{E}\|\nabla J(\theta_{k+1}) - g(\tau_{k+1}|\theta_{k+1})\|^2 \\
&\quad + 2(1-\beta_{k+1})^2\mathbb{E}\|g(\tau_{k+1}|\theta_{k+1}) - w(\tau_{k+1}|\theta_k,\theta_{k+1})g(\tau_{k+1}|\theta_k) - (\nabla J(\theta_{k+1}) - \nabla J(\theta_k))\|^2 \\
&\le (1-\beta_{k+1})\mathbb{E}\|\nabla f(\theta_k) - u_k\|^2 + 2\beta_{k+1}^2\sigma^2 + 2\underbrace{\mathbb{E}\|g(\tau_{k+1}|\theta_{k+1}) - w(\tau_{k+1}|\theta_k,\theta_{k+1})g(\tau_{k+1}|\theta_k)\|^2}_{=T_1},
\end{aligned} \tag{42}$$

where the forth equality holds by $\mathbb{E}_{\tau_{k+1}\sim p(\tau|\theta_{k+1})}[g(\tau_{k+1}|\theta_{k+1})] = \nabla J(\theta_{k+1})$ and $\mathbb{E}_{\tau_{k+1}\sim p(\tau|\theta_{k+1})}[g(\tau_{k+1}|\theta_{k+1}) - w(\tau_{k+1}|\theta_k,\theta_{k+1})g(\tau_{k+1}|\theta_k)] = \nabla J(\theta_{k+1}) - \nabla J(\theta_k)$; the second last inequality follows by Young's inequality; and the last inequality holds by Assumption 2, and the inequality $\mathbb{E}\|\zeta - \mathbb{E}[\zeta]\|^2 = \mathbb{E}\|\zeta\|^2 - (\mathbb{E}[\zeta])^2 \le \mathbb{E}\|\zeta\|^2$, and $0 < \beta_{k+1} \le 1$.

Next, we give an upper bound of the term $T_1$ as follows:

$$
\begin{aligned}
T_1 &= \mathbb{E}\|g(\tau_{k+1}|\theta_{k+1}) - w(\tau_{k+1}|\theta_k,\theta_{k+1})g(\tau_{k+1}|\theta_k)\|^2 \\
&= \mathbb{E}\|g(\tau_{k+1}|\theta_{k+1}) - g(\tau_{k+1}|\theta_k) + g(\tau_{k+1}|\theta_k) - w(\tau_{k+1}|\theta_k,\theta_{k+1})g(\tau_{k+1}|\theta_k)\|^2 \\
&\le 2\mathbb{E}\|g(\tau_{k+1}|\theta_{k+1}) - g(\tau_{k+1}|\theta_k)\|^2 + 2\mathbb{E}\|(1 - w(\tau_{k+1}|\theta_k,\theta_{k+1}))g(\tau_{k+1}|\theta_k)\|^2 \\
&\le 2L^2\|\theta_{k+1} - \theta_k\|^2 + 2G^2\mathbb{E}\|1 - w(\tau_{k+1}|\theta_k,\theta_{k+1})\|^2 \\
&= 2L^2\|\theta_{k+1} - \theta_k\|^2 + 2G^2\mathbb{V}\big(w(\tau_{k+1}|\theta_k,\theta_{k+1})\big) \\
&\le 2(L^2 + 2G^2C_w^2)\|\theta_{k+1} - \theta_k\|^2,
\end{aligned}
\tag{43}
$$

where the second inequality holds by Lemma 1, and the third equality holds by Lemma 5, and the last inequality follows by Lemma 2.

Combining the inequalities (42) with (43), let $\hat{L}^2 = L^2 + 2G^2C_w^2$, we have

$$
\begin{aligned}
\mathbb{E}\|\nabla f(\theta_{k+1}) - u_{k+1}\|^2 &\le (1 - \beta_{k+1})\mathbb{E}\|\nabla f(\theta_k) - u_k\|^2 + 2\beta_{k+1}^2\sigma^2 + 4\hat{L}^2\|\theta_{k+1} - \theta_k\|^2 \\
&= (1 - \beta_{k+1})\mathbb{E}\|\nabla f(\theta_k) - u_k\|^2 + 2\beta_{k+1}^2\sigma^2 + 4\hat{L}^2\eta_k^2\|\tilde{\theta}_{k+1} - \theta_k\|^2.
\end{aligned}
$$

$\square$

**Theorem 4.** *Suppose the sequence $\{\theta_k\}_{k=1}^K$ be generated from Algorithm 2. Let $\eta_k = \frac{b}{(m+k)^{1/3}}$ for all $k \ge 0$, $0 < \lambda \le \frac{\nu m^{1/3}}{5\hat{L}b}$, $b > 0$, $\frac{2}{3b^3} + \frac{20\hat{L}^2\lambda^2}{\nu^2} \le c \le \frac{m^{2/3}}{b^2}$ and $m \ge \max\big(2, b^3, (cb)^3, (\frac{5}{6b})^{2/3}\big)$, we have*

$$
\frac{1}{K}\sum_{k=1}^K \mathbb{E}\|\mathcal{B}_{\lambda,\langle\nabla f(\theta_k),\theta\rangle}^{\psi_k}(\theta_k)\| \le \frac{2\sqrt{2M'}m^{1/6}}{K^{1/2}} + \frac{2\sqrt{2M'}}{K^{1/3}},
\tag{44}
$$

*where $M' = \frac{J^* - J(\theta_1)}{b\nu\lambda} + \frac{m^{1/3}\sigma^2}{16b^2\hat{L}^2\lambda^2} + \frac{c^2\sigma^2b^2}{8\hat{L}^2\lambda^2}$ and $\hat{L}^2 = L^2 + 2G^2C_w^2$.*

*Proof.* Since $\eta_k = \frac{b}{(m+k)^{1/3}}$ on $k$ is decreasing and $m \ge b^3$, we have $\eta_k \le \eta_0 = \frac{b}{m^{1/3}} \le 1$. Due to $\hat{L} = \sqrt{L^2 + 2G^2C_w^2} \ge L$, we have $0 < \lambda \le \frac{\nu m^{1/3}}{5\hat{L}b} \le \frac{\nu m^{1/3}}{2Lb} = \frac{\nu}{2L\eta_0} \le \frac{\nu}{2L\eta_k}$ for any $k \ge 0$. Consider $0 < \eta_k \le 1$ and $m \ge (cb)^3$, we have $\beta_{k+1} = c\eta_k^2 \le \frac{cb^2}{m^{2/3}} \le 1$. At the same time, we have $c \le \frac{m^{2/3}}{b^2}$. According to Lemma 8, we have

$$
\begin{aligned}
&\frac{1}{\eta_k}\mathbb{E}\|\nabla f(\theta_{k+1}) - u_{k+1}\|^2 - \frac{1}{\eta_{k-1}}\mathbb{E}\|\nabla f(\theta_k) - u_k\|^2 \\
&\le \Big(\frac{1 - \beta_{k+1}}{\eta_k} - \frac{1}{\eta_{k-1}}\Big)\mathbb{E}\|\nabla f(\theta_k) - u_k\|^2 + \frac{2\beta_{k+1}^2\sigma^2}{\eta_k} + 4\hat{L}^2\eta_k\|\tilde{\theta}_{k+1} - \theta_k\|^2 \\
&= \Big(\frac{1}{\eta_k} - \frac{1}{\eta_{k-1}} - c\eta_k\Big)\mathbb{E}\|\nabla f(\theta_k) - u_k\|^2 + 2c^2\eta_k^3\sigma^2 + 4\hat{L}^2\eta_k\|\tilde{\theta}_{k+1} - \theta_k\|^2 \\
&= \Big(\frac{1}{b}\big((m+k)^{\frac{1}{3}} - (m+k-1)^{\frac{1}{3}}\big) - c\eta_k\Big)\mathbb{E}\|\nabla f(\theta_k) - u_k\|^2 + 2c^2\eta_k^3\sigma^2 + 4\hat{L}^2\eta_k\|\tilde{\theta}_{k+1} - \theta_k\|^2 \\
&\le \Big(\frac{2}{3b^3}\eta_k - c\eta_k\Big)\mathbb{E}\|\nabla f(\theta_k) - u_k\|^2 + 2c^2\eta_k^3\sigma^2 + 4\hat{L}^2\eta_k\|\tilde{\theta}_{k+1} - \theta_k\|^2,
\end{aligned}
\tag{45}
$$

where the last inequality holds by the following inequality

$$
\begin{aligned}
(m+k)^{\frac{1}{3}} - (m+k-1)^{\frac{1}{3}} &\le \frac{1}{3(m+k-1)^{2/3}} \le \frac{1}{3\big(m/2+k\big)^{2/3}} \\
&\le \frac{2^{2/3}}{3(m+k)^{2/3}} = \frac{2^{2/3}}{3b^2}\frac{b^2}{(m+k)^{2/3}} = \frac{2^{2/3}}{3b^2}\eta_k^2 \le \frac{2}{3b^2}\eta_k,
\end{aligned}
\tag{46}
$$

where the first inequality holds by the concavity of function $f(x) = x^{1/3}$, *i.e.*, $(x + y)^{1/3} \leq x^{1/3} + \frac{y}{3x^{2/3}}$; the second inequality is due to $m \geq 2$, and the last inequality is due to $0 < \eta_k \leq 1$. Let $c \geq \frac{2}{3b^3} + \frac{20\hat{L}^2\lambda^2}{\nu^2}$, we have

$$\frac{1}{\eta_k}\mathbb{E}\|\nabla f(\theta_{k+1}) - u_{k+1}\|^2 - \frac{1}{\eta_{k-1}}\mathbb{E}\|\nabla f(\theta_k) - u_k\|^2$$
$$\leq -\frac{20\hat{L}^2\lambda^2}{\nu^2}\eta_k\mathbb{E}\|\nabla f(\theta_k) - u_k\|^2 + 2c^2\eta_k^3\sigma^2 + 4\hat{L}^2\eta_k\|\tilde{\theta}_{k+1} - \theta_k\|^2. \tag{47}$$

Here we simultaneously consider $c \geq \frac{2}{3b^3} + \frac{20\hat{L}^2\lambda^2}{\nu^2}$, $c \leq \frac{m^{2/3}}{b^2}$ and $0 < \lambda \leq \frac{\nu m^{1/3}}{5\hat{L}b}$, we have

$$\frac{2}{3b^3} + \frac{20\hat{L}^2\lambda^2}{\nu^2} \leq \frac{2}{3b^3} + \frac{20\hat{L}^2}{\nu^2}\frac{\nu^2 m^{2/3}}{25\hat{L}^2b^2} = \frac{2}{3b^3} + \frac{4m^{2/3}}{5b^2} \leq \frac{m^{2/3}}{b^2}. \tag{48}$$

Then we have $m \geq (\frac{5}{6b})^{2/3}$.

Next we define a *Lyapunov* function $\Omega_k = \mathbb{E}[f(\theta_k) + \frac{\nu}{16\hat{L}^2\lambda\eta_{k-1}}\|\nabla f(\theta_k) - u_k\|^2]$ for any $k \geq 1$. According to Lemma 6, we have

$$\Omega_{k+1} - \Omega_k = f(\theta_{k+1}) - f(\theta_k) + \frac{\nu}{16\hat{L}^2\lambda}\left(\frac{1}{\eta_k}\mathbb{E}\|\nabla f(\theta_{k+1}) - u_{k+1}\|^2 - \frac{1}{\eta_{k-1}}\mathbb{E}\|\nabla f(\theta_k) - u_k\|^2\right)$$
$$\leq \frac{\eta_k\lambda}{\nu}\|\nabla f(\theta_k) - u_k\|^2 - \frac{\nu\eta_k}{2\lambda}\|\tilde{\theta}_{k+1} - \theta_k\|^2 - \frac{5\lambda\eta_k}{4\nu}\mathbb{E}\|\nabla f(\theta_k) - u_k\|^2$$
$$+ \frac{\nu\eta_k}{4\lambda}\mathbb{E}\|\tilde{\theta}_{k+1} - \theta_k\|^2 + \frac{\nu c^2\eta_k^3\sigma^2}{8\hat{L}^2\lambda}$$
$$\leq -\frac{\lambda\eta_k}{4\nu}\mathbb{E}\|\nabla f(\theta_k) - u_k\|^2 - \frac{\nu\eta_k}{4\lambda}\mathbb{E}\|\tilde{\theta}_{k+1} - \theta_k\|^2 + \frac{\nu c^2\eta_k^3\sigma^2}{8\hat{L}^2\lambda}, \tag{49}$$

where the first inequality is due to the above inequality (47). Thus, we can obtain

$$\frac{\lambda\eta_k}{4\nu}\mathbb{E}\|\nabla f(\theta_k) - u_k\|^2 + \frac{\nu\eta_k}{4\lambda}\mathbb{E}\|\tilde{\theta}_{k+1} - \theta_k\|^2 \leq \Omega_k - \Omega_{k+1} + \frac{\nu c^2\eta_k^3\sigma^2}{8\hat{L}^2\lambda}. \tag{50}$$

Taking average over $k = 1, 2, \cdots, K$ on both sides of (50), we have

$$\frac{1}{T}\sum_{k=1}^K\mathbb{E}\Big[\frac{\lambda\eta_k}{4\nu}\mathbb{E}\|\nabla f(\theta_k) - u_k\|^2 + \frac{\nu\eta_k}{4\lambda}\mathbb{E}\|\tilde{\theta}_{k+1} - \theta_k\|^2\Big]$$
$$\leq \frac{f(\theta_1) - f(\theta_{K+1})}{K} + \frac{\nu\|\nabla f(\theta_1) - u_1\|^2}{16\hat{L}^2\eta_0\lambda K} - \frac{\nu\|\nabla f(\theta_{K+1}) - u_{K+1}\|^2}{16\hat{L}^2\eta_K\lambda K} + \frac{1}{K}\sum_{k=1}^K\frac{\nu c^2\eta_k^3\sigma^2}{8\hat{L}^2\lambda}$$
$$\leq \frac{J(\theta_{K+1}) - J(\theta_1)}{K} + \frac{\nu\sigma^2}{16\hat{L}^2\eta_0\lambda K} + \frac{1}{K}\sum_{k=1}^K\frac{\nu c^2\eta_k^3\sigma^2}{8\hat{L}^2\lambda}$$
$$\leq \frac{J^* - J(\theta_1)}{K} + \frac{\nu\sigma^2}{16\hat{L}^2\eta_0\lambda K} + \frac{1}{K}\sum_{k=1}^K\frac{\nu c^2\eta_k^3\sigma^2}{8\hat{L}^2\lambda}, \tag{51}$$

where the second inequality is due to $u_1 = -g(\tau_1|\theta_1)$, $\nabla f(\theta_1) = -\nabla J(\theta_1)$ and Assumption 1, and the last inequality holds by Assumption 2. Since $\eta_k$ is decreasing, *i.e.*, $\eta_K^{-1} \geq \eta_k^{-1}$ for any

$0 < k \le K$, we have

$$\frac{1}{K}\sum_{k=1}^{K}\mathbb{E}\big[\frac{1}{4\nu^2}\mathbb{E}\|\nabla f(\theta_k) - u_k\|^2 + \frac{1}{4\lambda^2}\mathbb{E}\|\tilde{\theta}_{k+1} - \theta_k\|^2\big]$$

$$\le \frac{J^* - J(\theta_1)}{K\nu\lambda\eta_K} + \frac{\sigma^2}{16\hat{L}^2\eta_K\eta_0\lambda^2 K} + \frac{1}{K\lambda\eta_K}\sum_{k=1}^{K}\frac{c^2\eta_k^3\sigma^2}{8\hat{L}^2\lambda}$$

$$\le \frac{J^* - J(\theta_1)}{K\nu\lambda\eta_K} + \frac{m^{1/3}\sigma^2}{16b\hat{L}^2\lambda^2\eta_K K} + \frac{c^2\sigma^2}{8\hat{L}^2\lambda^2 K\eta_K}\int_1^K \frac{b^3}{m+k}dk$$

$$\le \frac{J^* - J(\theta_1)}{K\lambda\eta_K} + \frac{m^{1/3}\sigma^2}{16b\hat{L}^2\lambda^2\eta_K K} + \frac{c^2\sigma^2 b^3}{8\hat{L}^2\lambda^2 K\eta_K}\ln(m+K)$$

$$= \left(\frac{J^* - J(\theta_1)}{b\nu\lambda} + \frac{m^{1/3}\sigma^2}{16b^2\hat{L}^2\lambda^2} + \frac{c^2\sigma^2 b^2}{8\hat{L}^2\lambda^2}\right)\frac{(m+K)^{1/3}}{K}, \tag{52}$$

where the second inequality holds by $\sum_{k=1}^{K}\eta_k^3 dk \le \int_1^K \eta_k^3 dk = b^3\int_1^K (m+k)^{-1}dk$.

Let $M' = \frac{J^* - J(\theta_1)}{b\nu\lambda} + \frac{m^{1/3}\sigma^2}{16b^2\hat{L}^2\lambda^2} + \frac{c^2\sigma^2 b^2}{8\hat{L}^2\lambda^2}$, the above inequality (52) reduces to

$$\frac{1}{K}\sum_{k=1}^{K}\mathbb{E}\big[\frac{1}{4\nu^2}\|\nabla f(\theta_k) - u_k\|^2 + \frac{1}{4\lambda^2}\|\tilde{\theta}_{k+1} - \theta_k\|^2\big] \le \frac{M'}{K}(m+K)^{1/3}. \tag{53}$$

According to Jensen's inequality, we have

$$\frac{1}{K}\sum_{k=1}^{K}\mathbb{E}\big[\frac{1}{2\nu}\|\nabla f(\theta_k) - u_k\| + \frac{1}{2\lambda}\|\tilde{\theta}_{k+1} - \theta_k\|\big]$$

$$\le \big(\frac{2}{K}\sum_{k=1}^{K}\mathbb{E}\big[\frac{1}{4\nu^2}\|\nabla f(\theta_k) - u_k\|^2 + \frac{1}{4\lambda^2}\|\tilde{\theta}_{k+1} - \theta_k\|^2\big]\big)^{1/2}$$

$$\le \frac{\sqrt{2M'}}{K^{1/2}}(m+K)^{1/6} \le \frac{\sqrt{2M'}m^{1/6}}{K^{1/2}} + \frac{\sqrt{2M'}}{K^{1/3}}, \tag{54}$$

where the last inequality is due to the inequality $(a+b)^{1/6} \le a^{1/6} + b^{1/6}$ for all $a, b \ge 1$. Thus we have

$$\frac{1}{K}\sum_{k=1}^{K}\mathbb{E}\big[\frac{1}{\nu}\|\nabla f(\theta_k) - u_k\| + \frac{1}{\lambda}\|\tilde{\theta}_{k+1} - \theta_k\|\big] \le \frac{2\sqrt{2M'}m^{1/6}}{K^{1/2}} + \frac{2\sqrt{2M'}}{K^{1/3}}. \tag{55}$$

Then by using the above inequality (40), we can obtain

$$\frac{1}{K}\sum_{k=1}^{K}\mathbb{E}\|\mathcal{B}_{\lambda,\langle\nabla f(\theta_k),\theta\rangle}^{\psi_k}(\theta_k)\| \le \frac{2\sqrt{2M'}m^{1/6}}{K^{1/2}} + \frac{2\sqrt{2M'}}{K^{1/3}}. \tag{56}$$

$\square$

# B  ACTOR-CRITIC STYLE BGPO AND VR-BGPO ALGORITHMS

In the experiments, we use the advantage-based policy gradient estimator:

$$g(\tau|\theta) = \sum_{t=0}^{H-1}\nabla\log\pi_\theta(a_t|s_t)\hat{A}^{\pi_\theta}(s_t, a_t), \tag{57}$$

---

**Algorithm 3** BGPO Algorithm (Actor-Critic Style)

---

1: **Input:** Total iteration $K$, tuning parameters $\{\lambda, b, m, c\}$ and mirror mappings $\{\psi_k\}_{k=1}^{K}$ are $\nu$-strongly convex functions;
2: **Initialize:** $\theta_1 \in \Theta$, $\theta_1^v \in \Theta^v$ and sample a trajectory $\tau_1$ from $p(\tau|\theta_1)$, and compute $u_1 = -g(\tau_1|\theta_1)$;
3: **for** $k = 1, 2, \ldots, K$ **do**
4:   *# Update the policy network*
5:   Update $\tilde{\theta}_{k+1} = \arg\min_{\theta \in \Theta} \left\{ \langle u_k, \theta \rangle + \frac{1}{\lambda} D_{\psi_k}(\theta, \theta_k) \right\}$;
6:   Update $\theta_{k+1} = \theta_k + \eta_k(\tilde{\theta}_{k+1} - \theta_k)$ with $\eta_k = \frac{b}{(m+k)^{1/2}}$;
7:   *# Update the value network*
8:   Update $\theta_{k+1}^v$ by solving the subproblem (58);
9:   *# Sample a new trajectory and compute policy gradients*
10:   Sample a trajectory $\tau_{k+1}$ from $p(\tau|\theta_{k+1})$, and compute $u_{k+1} = -\beta_{k+1} g(\tau_{k+1}|\theta_{k+1}) + (1 - \beta_{k+1}) u_k$ with $\beta_{k+1} = c\eta_k$;
11: **end for**
12: **Output:** $\theta_\zeta$ chosen uniformly random from $\{\theta_k\}_{k=1}^{K}$.

---

**Algorithm 4** VR-BGPO Algorithm (Actor-Critic Style)

---

1: **Input:** Total iteration $K$, tuning parameters $\{\lambda, b, m, c\}$ and mirror mappings $\{\psi_k\}_{k=1}^{K}$ are $\nu$-strongly convex functions;
2: **Initialize:** $\theta_1 \in \Theta$, $\theta_1^v \in \Theta^v$ and sample a trajectory $\tau_1$ from $p(\tau|\theta_1)$, and compute $u_1 = -g(\tau_1|\theta_1)$;
3: **for** $k = 1, 2, \ldots, K$ **do**
4:   *# Update the policy network*
5:   Update $\tilde{\theta}_{k+1} = \arg\min_{\theta \in \Theta} \left\{ \langle u_k, \theta \rangle + \frac{1}{\lambda} D_{\psi_k}(\theta, \theta_k) \right\}$;
6:   Update $\theta_{k+1} = \theta_k + \eta_k(\tilde{\theta}_{k+1} - \theta_k)$ with $\eta_k = \frac{b}{(m+k)^{1/3}}$;
7:   *# Update the value network*
8:   Update $\theta_{k+1}^v$ by solving the subproblem (58);
9:   *# Sample a new trajectory and compute policy gradients*
10:   Sample a trajectory $\tau_{k+1}$ from $p(\tau|\theta_{k+1})$, and compute $u_{k+1} = -\beta_{k+1} g(\tau_{k+1}|\theta_{k+1}) + (1 - \beta_{k+1}) \left[ u_k - g(\tau_{k+1}|\theta_k) + w(\tau_{k+1}|\theta_k, \theta_{k+1}) g(\tau_{k+1}|\theta_k) \right]$ with $\beta_{k+1} = c\eta_k^2$;
11: **end for**
12: **Output:** $\theta_\zeta$ chosen uniformly random from $\{\theta_k\}_{k=1}^{K}$.

---

where $\theta (\in \Theta \subseteq \mathbb{R}^d)$ denotes parameters of the policy network, and $\hat{A}^{\pi_\theta}(s, a)$ is an estimator of the advantage function $A^{\pi_\theta}(s, a)$. In using advantage-based policy gradient, we also need the state-value function $V^{\pi_\theta}(s)$. Here, we use a value network $V_{\theta^v}(s)$ to approximate the state-value function $V^{\pi_\theta}(s)$. Specifically, we solve the following problem to obtain the value network:

$$\min_{\theta^v \in \Theta^v} \mathcal{L}(\theta^v) := \sum_{t=0}^{H-1} \left( V_{\theta^v}(s_t) - \hat{V}^{\pi_\theta}(s_t) \right)^2, \tag{58}$$

where $\theta^v (\in \Theta^v \subseteq \mathbb{R}^{d_v})$ denotes parameters of the value network, and $\hat{V}^{\pi_\theta}(s)$ is an estimator of the state-value function $V^{\pi_\theta}(s)$, which is obtained by the GAE Schulman et al. (2016). Then we use the GAE to estimate $\hat{A}^{\pi_\theta}$ based on value network $V_{\theta^v}$. We describe the actor-critic style BGPO and VR-BGPO algorithms in Algorithm 3 and Algorithm 4, respectively.

## C  DETAILED SETUP OF EXPERIMENTAL ENVIRONMENTS AND HYPER-PARAMETERS

In this section, we provide the detailed setup of experimental environments and hyper-parameters. We first provide the detailed setup of our experiments in Tab. 2 and Tab. 3. We use ADAM optimizer to optimize value functions for all methods and settings, which is a common practice. The impor-

| Environments | CartPole-v1 | Acrobat-v1 | MountainCar-v0 |
|---|---|---|---|
| Horizon | 100 | 500 | 500 |
| Value function Network sizes | $32 \times 32$ | $32 \times 32$ | $32 \times 32$ |
| Policy network sizes | $8 \times 8$ | $8 \times 8$ | $64 \times 64$ |
| Number of timesteps | $5 \times 10^5$ | $5 \times 10^6$ | $7.5 \times 10^6$ |
| Batchsize | 50 | 100 | 100 |
| VR-BGPO/BGPO $\{b, m, c\}$ | $\{1.5, 2.0, 25\}$ | $\{1.5, 2.0, 25\}$ | $\{1.5, 2.0, 25\}$ |
| BGPO-$l_p$ $\{\lambda_{p=1.5}, \lambda_{p=2.0}, \lambda_{p=3.0}\}$ | $\{0.0064, 0.0016, 0.0008\}$ | $\{0.016, 0.004, 0.001\}$ | $\{0.016, 0.004, 0.001\}$ |
| BGPO-Diag/VR-BGPO-Diag $\lambda$ | $1 \times 10^{-3}$ | $1 \times 10^{-3}$ | $1 \times 10^{-3}$ |
| Value function learning rate | $2.5 \times 10^{-3}$ | $2.5 \times 10^{-3}$ | $2.5 \times 10^{-3}$ |

Table 2: Setups of environments and hyper-parameters for experiments in section 6.2 and section 6.3. The learning rate of value functions are the same for all methods.

| Environments | Pendulum-v2 | DoublePendulum-v2 | Walker2d-v2 | Swimmer-v2 | Reacher-v2 | HalfCheetah-v2 |
|---|---|---|---|---|---|---|
| Horizon | 500 | 500 | 500 | 500 | 500 | 500 |
| Value function Network sizes | $32 \times 32$ | $32 \times 32$ | $32 \times 32$ | $32 \times 32$ | $32 \times 32$ | $32 \times 32$ |
| Policy network sizes | $64 \times 64$ | $64 \times 64$ | $64 \times 64$ | $64 \times 64$ | $64 \times 64$ | $64 \times 64$ |
| Number of timesteps | $5 \times 10^6$ | $5 \times 10^6$ | $1 \times 10^7$ | $1 \times 10^7$ | $1 \times 10^7$ | $1 \times 10^7$ |
| Batchsize | 100 | 100 | 100 | 100 | 100 | 100 |
| VR-BGPO $\{b, m, c\}$ | $\{1.50, 2.0, 25\}$ | $\{1.50, 2.0, 25\}$ | $\{1.50, 2.0, 25\}$ | $\{1.50, 2.0, 25\}$ | $\{1.50, 2.0, 25\}$ | $\{1.50, 2.0, 25\}$ |
| VR-BGPO $\lambda$ | $1 \times 10^{-2}$ | $1 \times 10^{-2}$ | $1 \times 10^{-2}$ | $5 \times 10^{-4}$ | $5 \times 10^{-4}$ | $5 \times 10^{-4}$ |
| TRPO/PPO learning rate | $2.5 \times 10^{-3}$ | $2.5 \times 10^{-3}$ | $2.5 \times 10^{-3}$ | $2.5 \times 10^{-3}$ | $2.5 \times 10^{-3}$ | $2.5 \times 10^{-3}$ |
| MDPO learning rate | $3 \times 10^{-3}$ | $3 \times 10^{-3}$ | $3 \times 10^{-3}$ | $3 \times 10^{-3}$ | $3 \times 10^{-3}$ | $3 \times 10^{-3}$ |
| VRMPO learning rate | $5 \times 10^{-3}$ | $3 \times 10^{-4}$ | $1 \times 10^{-2}$ | $2 \times 10^{-4}$ | $5 \times 10^{-5}$ | $5 \times 10^{-5}$ |
| Value function learning rate | $2.5 \times 10^{-3}$ | $2.5 \times 10^{-3}$ | $2.5 \times 10^{-3}$ | $2.5 \times 10^{-3}$ | $2.5 \times 10^{-3}$ | $2.5 \times 10^{-3}$ |

Table 3: Setups of environments and hyper-parameters for experiments in section 6.4. The learning rate of value functions are the same for all methods.

tance sampling weight used for VR-BGPO algorithm is clipped within $[0.5, 1.5]$. The momentum term $\beta_k$ is set to be less or equal than one ($\beta_k = \min(\beta_k, 1.0)$ ) through the whole training process.

BGPO and VR-BGPO algorithms involve 4 hyper-parameters $\{\lambda, b, m, c\}$, which may bring additional efforts for hyper-parameter tuning. However, the actual hyper-parameter tuning is not so hard, and we only use one set of $\{b, m, c\}$ for 9 environments. The strategy of hyper-parameter tuning is to separate the four hyper-parameters into two parts. The first part is $\{b, m, c\}$, which mainly decide when the momentum term $\beta_k$ actually affects ($\beta_k < 1.0$) updates. The second part, $\lambda$, only affects how fast the policy is learning. To further reduce the complexity of hyper-parameter tuning, we always set $m = 2$. By grouping hyper-parameters, we only consider $\lambda$ and how $\beta_k$ changes, which largely simplifies the process of hyper-parameter tuning.

