# OpenReview forum: "Bregman Gradient Policy Optimization"
_ICLR.cc/2022/Conference — ICLR 2022 Poster_

### Official Review · Reviewer_pxbY · 2021-10-31

**Correctness:** 4
**Technical Novelty And Significance:** 3
**Empirical Novelty And Significance:** Not applicable
**Recommendation:** 8
**Confidence:** 5

**Details Of Ethics Concerns:**

No ethics issue.

**Main Review:**

The paper discusses the momentum and STORM version of the mirror descent PG method, which is a relatively new result. The analysis and the presentation of the results are clear and well-organized. Although the result can be expected since both STORM and mirror descent, as well as their numerous variants, are well-studied, the discovery of this paper does imply the convergence of several important special cases of the PG method, including the NPG and Super-Adam version of PG. So I think this is a good paper.

**Summary Of The Paper:**

The authors consider the optimization problem of an MDP. They designed two policy gradient algorithms based on the mirror descent method, which are named BGPO and VR-BGPO. The BGPO algorithm is a momentum mirror descent method that finds an $\epsilon$-stationary policy with $O(\epsilon^{-4})$ samples. The VR-BGPO algorithm is a STORM-type variance reduced mirror descent method that finds an $\epsilon$-stationary policy with $O(\epsilon^{-3})$ samples. The analysis is nicely organized and the authors also provide a couple of experiments to verify their theoretical findings.

**Summary Of The Review:**

The main comments are provided in the main review part. Here I will add a few comments.

1. The authors have used "mirror descend" several times in the paper, please unify terminology and use "mirror descent".

2. Because the second algorithm proposed by the author is a STORM type variance reduced mirror descent PG method, the authors should also mention the work by Yuhao Ding Junzi Zhang & Javad Lavaei: \emph{On the Global Convergence of Momentum-based Policy Gradient}. The algorithms of this paper also apply the STORM technique.  For the same reason, the following work by Nhan H. Pham et al. \emph{A Hybrid Proximal Stochastic Policy Gradient Algorithm} should also be cited.

3. Regarding equation (4), the author should mention that the objective function with a horizon $H$, there is a truncation error of $\frac{\gamma^H}{1-\gamma}$ compared to the original infinite-horizon MDP.

4. Regarding Assumption 2 and Assumption 4, the author should mention that both of them are satisfied automatically given Assumption 1 and the fact that all the rewards $r(s,a)$ are bounded.


5. Regarding Assumption 3, the authors should also mention that the importance weights can be bounded theoretically (instead of an uncheckable assumption) by using the truncated gradient step instead of a gradient step. See the work of Junyu Zhang et al. \emph{On the convergence and sample efficiency of variance-reduced policy gradient method}. The importance weights actually as some nice properties,   for soft-max policy it is bounded by $e^{c||\theta_1-\theta_2||}$, in Gaussian policy it is bounded as $e^{c||\theta_1-\theta_2||^2}$. So a simple truncated update is enough to control this value.

6. There are still a few typos/gramma errors in the paper, for example, in the first row of contribution (a), the "based the" should be "based on the". The authors should carefully check the spelling and gramma in the revision.

7. Regarding the experiments, the authors compared with several different algorithms. However, there is a small issue w.r.t. the selected algorithms.

VR-BGPO: variance reduction (VR) + mirror descent (MD)

VR-MDPO: VR + MD

MDPO: VR only

TPO & PPO: no VR, no MD

From these comparisons, I'm not able to see if the MD really works. The authors should also compare with algorithms such as Prox-HSPGA, (A hybrid stochastic policy gradient algorithm for reinforcement learning) which applies STORM technique while not using mirror descent.

---

> ### Author Response · Authors · 2021-11-12
> **Responses to Reviewer pxbY**
>
> Thanks so much for your positive comments about our paper.
>
> R1： Thanks for pointing this typo. In our new version, we will correct it.
>
> R2:  Thanks for your suggestion. In our new version, we will mention these papers and cite them.
>
> R3-5: Thanks for your suggestion, we will mention them in our new version.
>
> R6: Thanks for pointing these typos. We will correct them in our new version.
>
> R7：Thanks for your suggestion. In our new version, we will compare our BGPO **(no VR, MD only)** with TPO $\&$ PPO **( no VR, no MD)**, and  compare our BGPO and VR-BGPO methods with **Prox-HSPGA**.
>
> **********************************************************************************************
>
> We have uploaded our new version. In our new version, we have used blue for the main revisions. At the same time, we have added two comparison methods (i.e., BGPO and Prox-HSPGA) in Figure 3. Due to the recent deadline of CVPR, the computing resources of our lab are occupied. So the results of two environments (InvertedPendulum-v2 and HalfCheetah-v2) have not yet come out. We will add these two results in the final version.
>
> Thanks again for the great effort in reviewing our manuscript and for helping us improve its quality.

---

### Official Review · Reviewer_mQZe · 2021-11-04

**Correctness:** 4
**Technical Novelty And Significance:** 1
**Empirical Novelty And Significance:** 1
**Recommendation:** 6
**Confidence:** 4

**Main Review:**

My concern about this paper mainly lies in its lack of novelty both in the algorithm design and the theoretical analysis. First of all, mirro policy gradient with Bregman divergence regularizer was already proposed in other papers. For example, the algorithm proposed in the current work is almost the same as that in the following paper:
Yang L, Zheng G, Zhang H, Zhang Y, Zheng Q, Wen J, Pan G. Policy optimization with stochastic mirror descent. arXiv preprint arXiv:1906.10462. 2019 Jun 25.

The difference of the current paper with the above one lies in the variance reduction techniques. Yang et al uses the SPIDER estimator, while the current paper uses the STORM estimator proposed in Cutkosky and Orabona (2019). Moreover, even the adaptation of STORM from nonconvex optimization to policy optimization was already studied in a recent paper: Feihu Huang, Shangqian Gao, Jian Pei, and Heng Huang. Momentum-based policy gradient methods (2020). Therefore, combining these methods with a mirror descent update do not seem to have enough contributions for the publishment of this paper.


==after the author's response==
I thank the authors for their detailed response on the differences of the submission from closely related work. Given these comments and discussions, I am willing to raise my score to accept this paper.

**Summary Of The Paper:**

This paper studies the convergence of policy gradient algorithms with constraints. They modify the vanilla policy gradient with Bregman divergence as a regularizer. The authors also propose a new variance reduced policy gradient methods based on the STORM estimator in nonconvex optimization.




**Summary Of The Review:**

This paper has limited novelty in both algorithm design and theoretical analysis. It is also highly similar to existing papers from several perspectives.

---

> ### Author Response · Authors · 2021-11-10
> **Rebuttal to Reviewer mQZe**
>
> Our methods (i.e., BGPO and VR-BGPO) are totally different from the methods (i.,e, MPO and VRMPO) of (Yang et al., 2019, https://openreview.net/forum?id=SkxpDT4YvS,  https://arxiv.org/pdf/1906.10462.pdf).
> We **first** compare our non-variance-reduced method (i.e., BGPO) with the non-variance-reduced method (i.e., MPO ) of (Yang et al., 2019):
>
> 1) Our BGPO algorithm uses a useful **momentum iteration**: $ \theta_{k+1} = \theta_k + \eta_k(\tilde{\theta}_{k+1}-\theta_k)$ at the step 5 of our Algorithm 1. While MPO algorithm of (Yang et al., 2019) does not use a similar iteration.
>
> 2) Our BGPO uses a momentum-based policy gradient estimator $u_{k+1} = -\beta_{k+1} g(\tau_{k+1}|\theta_{k+1}) + (1-\beta_{k+1})u_k$ at the step 6 of our Algorithm 1, which includes stochastic policy gradients $( g(\tau_i  | \theta_i ))_{i=1}^{k+1}$ .
>
> While MPO of (Yang et al., 2019) uses a policy gradient estimator $G_k = \frac{1}{k} g_k + (1- \frac{1}{k}) g_{k-1}$ at (22) of Algorithm 1 in (Yang et al., 2019), which only includes two stochastic policy gradients
> $(g(\tau_{k}|\theta_{k}), g(\tau_{k-1}|\theta_{k-1}))$. Clearly, the policy gradient estimator used in our BGPO is totally different from the policy gradient estimator used in MPO of (Yang et al., 2019).
>
> 3) We provide a solid convergence analysis for our BGPO, and prove that our BGPO has a sample complexity of $O(\epsilon^{-4})$ for finding an $\epsilon $-accurate stationary policy only requiring one trajectory at each iteration. While the convergence results of MPO given in (Yang et al., 2019) are **incorrect** (Please see the openreview of **Paper613 AnonReviewer2** in https://openreview.net/forum?id=SkxpDT4YvS )，e.g., a key equality (40) at page 16 of (Yang et al., 2019) is not true.
>
> We **then** compare our variance-reduced method (i.e., VR-BGPO) with the variance-reduced method (i.e., VRMPO )  of (Yang et al., 2019):
>
> 1) Our VR-BGPO uses a useful **momentum iteration** $ \theta_{k+1} = \theta_k + \eta_k(\tilde{\theta}_{k+1}-\theta_k)$ at the step 5 of our Algorithm 2. While VRMPO of (Yang et al., 2019) does not use a similar iteration.
>
> 2) Our VR-BGPO uses the variance-reduced technique of STORM without relying on large batches. While VRMPO of (Yang et al., 2019) uses the variance-reduced technique of SPIDER relying on large batches. Clearly, the policy gradient estimator used in our VR-BGPO is totally different from the policy gradient estimator used in VRMPO of (Yang et al., 2019). At the same time, our VR-BGPO algorithm is a single-loop, while VRMPO algorithm is a double-loop.
>
> 3) We provide a solid convergence analysis for our VR-BGPO, and prove that our VR-BGPO has a sample complexity of $O(\epsilon^{-3})$ for finding an $\epsilon $-accurate stationary policy **only requiring one trajectory at each iteration**. Although  (Yang et al., 2019) also provided the convergence properties of VRMPO relying on large batches, the convergence results of  VRMPO given in (Yang et al., 2019) are **incorrect** (Please see the openreview of **Paper613 AnonReviewer2** in https://openreview.net/forum?id=SkxpDT4YvS )，e.g., the derivation in (55) at page 20 of (Yang et al., 2019) is not true.
>
> Our methods (i.e., BGPO and VR-BGPO) are very different from the MBPG method of (Huang et al., 2020, http://proceedings.mlr.press/v119/huang20a/huang20a.pdf).
>
> 1)  Our BGPO and VR-BGPO use a useful **momentum iteration** $ \theta_{k+1} = \theta_k + \eta_k(\tilde{\theta}_{k+1}-\theta_k)$ at the step 5. While MBPG of (Huang et al., 2020) does not use a similar iteration.
>
> 2) Our BGPO and VR-BGPO use a **constant learning rate**(i.e., $\lambda$) at the step 4 of our Algorithms 1 and 2. While MBPG only uses a **decreasing learning rate** (i.e., $\eta_t = \frac{k}{(m+\sum_{i=1}^tG^2_i)^{1/3}}$ or $\eta_t = \frac{k}{(m+t)^{1/3}}$) at step 10 of Algorithms 1-3 in (Huang et al., 2020) .
>
> 3) Our BGPO and VR-BGPO consider the **constrained** policy problem with convex set $\Theta$. While MBPGs of  (Huang et al., 2020) consider the **unconstrained** policy problem.
>
> 4) We study both the non-variance-reduced and variance-reduced policy gradient methods, i.e., BGPO and VR-BGPO. While  (Huang et al., 2020) only studies the variance-reduced policy gradient methods, i.e., MBPGs.
>
> 5) Since our BGPO/VR-BGPO and MBPG of  (Huang et al., 2020)  algorithms are different, clearly, our convergence analysis are very different from  (Huang et al., 2020). E.g., In our convergence analysis, the mirror descent iteration is more complex than a simple gradient ascent iteration in (Huang et al., 2020) .

---

### Official Review · Reviewer_s4n4 · 2021-11-04

**Correctness:** 3
**Technical Novelty And Significance:** 3
**Empirical Novelty And Significance:** 3
**Recommendation:** 8
**Confidence:** 3

**Main Review:**

In general, the paper is well written with some minor grammar issues. The developed BGPO and VR-BGPO are simple and easy to implement. The derived convergence results match the state-of-the-art one. Numerical results illustrate the effectiveness of the proposed algorithms. However, we still have some concerns as follows:

1) My understanding is that the proposed algorithm is pretty general. Why does the work only focus on policy gradient rather than general gradient? What is the unique property that has been used here to guarantee theoretical results compared to the traditional stochastic gradient methods?

2) Why do the compared algorithms shown in Table 3 need a large batch size (i.e., $\epsilon$ dependent) while the proposed one only requires constant batch size?

3) I agree that VR-BGPO is better than BGPO. But BGPO is not simulated in Figure 3?

Some other minor issues:

PGT is not explicitly definied.

**Summary Of The Paper:**

This work proposes a Bregman gradient policy optimization framework for RL. Two specific algorithms are proposed, which are BGPO and VR-BGPO, where VR-BGPO is an accelerated version of BGPO. The authors provide the convergence rates results for these two algorithms and show their efficiency through multiple numerical simulations.

**Summary Of The Review:**

Overall, I believe that there are some merits of this work regarding the simplicity of the implementation of the algorithms, theoretical justification of the convergence rates, and numerical instances of verifying the efficiency of BGPO and VR-BGPO.

---

> ### Author Response · Authors · 2021-11-12
> **Responses to Reviewer s4n4**
>
> Thanks so much for your positive comments about our paper.
>
> **R1**： Yes, you are right. In fact, we can use the general stochastic gradient instead of policy gradient in our BGPO and VR-BGPO methods, and then our methods will reduce to the **stochastic mirror decent + momentum** methods for the generic problems. For our VR-BGPO, we need to extra use an importance sampling weight in the policy gradient estimator ( at the step 6 of Algorithm 2 ) compared to the traditional stochastic gradient methods. We mainly focus on the reinforcement learning field, so we only consider policy gradient in our paper. Thanks very much for this good comment. In our new version, we will mention it in our contributions.
>
>
> **R2**:  Due to using **momentum-based policy gradient estimators** (the step 6 of Algorithms 1 and 2), our BGPO and VR-BGPO obtain the sample complexities of $O(\epsilon^{-4})$ and $O(\epsilon^{-3})$ respectively, for finding an $\epsilon $-accurate stationary policy **only requiring one trajectory at each iteration**.  However, due to **not** using momentum-based policy gradient estimators, the other policy gradient methods in Table 1 obtain their sample complexities by requiring large trajectories at each iteration. In addition,  the basic momentum policy gradient estimator (the step 6 of Algorithm 1) used in our BGPO is **very different** from the momentum variance-reduced policy gradient estimator (the step 6 of Algorithm 2) used in our VR-BGPO requiring an importance sampling weight.
> To the best of our knowledge, our BGPO method is the first work to use the basic momentum policy gradient estimator **without** requiring an importance sampling weight in policy gradient methods.
>
> **R3**:  Thanks for your suggestion. In our new version, we will add the BGPO in Figure 3. We are running the additional experiments.
>
> **R4**: Thanks for your suggestion. We will give the explicit definition of PGT.
>
>
> **********************************************************************************************
>
> We have uploaded our new version. In our new version, we have used blue for the main revisions. At the same time, we have added two comparison methods (i.e., BGPO and Prox-HSPGA) in Figure 3. Due to the recent deadline of CVPR, the computing resources of our lab are occupied. So the results of two environments (InvertedPendulum-v2 and HalfCheetah-v2) have not yet come out. We will add these two results in the final version.
>
> Thanks again for the great effort in reviewing our manuscript and for helping us improve its quality.

---

### Author Response · Authors · 2021-11-11
**Overall Responses (1)**

**Our contributions and novelties** of our paper are given as follows:

1) To the best of our knowledge, our BGPO and VR-BGPO methods are the first work to apply the momentum techniques to mirror-descent-type methods for regularized reinforcement learning. Specifically, our BGPO uses the momentum iteration (i.e., the step 5 of our Algorithm 1) to update parameter of policy, and applies the **basic momentum technique** to update the policy gradient estimator (i.e., the step 6 of our Algorithm 1). Our VR-BGPO also uses the momentum iteration (i.e., the step 5 of our Algorithm 2) to update parameter of policy, and applies the **variance-reduced momentum technique** of STROM to update the policy gradient estimator (i.e., the step 6 of our Algorithm 2).

2) We provide a solid convergence analysis for our BGPO and VR-BGPO methods. Specifically, we prove that our BGPO has a sample complexity of $O(\epsilon^{-4})$ for finding an $\epsilon $-accurate stationary policy only requiring one trajectory at each iteration. At the same time, we prove that our VR-BGPO has a sample complexity of $O(\epsilon^{-3})$ for finding an $\epsilon $-accurate stationary policy only requiring one trajectory at each iteration.

3) In fact, we can use the general stochastic gradient instead of policy gradient in our methods, and then our BGPO and VR-BGPO methods will reduce to the stochastic mirror decent + momentum methods. To the best of our knowledge, our paper is the first work to provide a solid convergence analysis for the stochastic mirror decent + momentum methods under non-convex setting.

**NOTE THAT**: Our methods (i.e., BGPO and VR-BGPO) are **totally different** from the methods (i.,e, MPO and VRMPO) of (Yang et al., 2019, https://openreview.net/forum?id=SkxpDT4YvS,  https://arxiv.org/pdf/1906.10462.pdf).
We **first** compare our non-variance-reduced method (i.e., BGPO) with the non-variance-reduced method (i.e., MPO ) of (Yang et al., 2019):

1) Our BGPO algorithm uses a useful **momentum iteration**: $ \theta_{k+1} = \theta_k + \eta_k(\tilde{\theta}_{k+1}-\theta_k)$ at the step 5 of our Algorithm 1. While MPO algorithm of (Yang et al., 2019) does **not** use a similar iteration.

2) Our BGPO uses a momentum-based policy gradient estimator $u_{k+1} = -\beta_{k+1} g(\tau_{k+1}|\theta_{k+1}) + (1-\beta_{k+1})u_k$ at the step 6 of our Algorithm 1, which includes stochastic policy gradients $\big(g(\tau_{i}|\theta_{i})\big)_{i=1}^{k+1}$ .

While MPO of (Yang et al., 2019) uses a policy gradient estimator $G_k = \frac{1}{k}g_k + (1-\frac{1}{k})g_{k-1}$ at (22) of Algorithm 1 in (Yang et al., 2019), which only includes two stochastic policy gradients $\big(g(\tau_{k}|\theta_{k}), g(\tau_{k-1}|\theta_{k-1})\big)$. Clearly, the policy gradient estimator used in our BGPO is **totally different** from the policy gradient estimator used in MPO of (Yang et al., 2019).

3) We provide a solid convergence analysis for our BGPO, and prove that our BGPO has a sample complexity of $O(\epsilon^{-4})$ for finding an $\epsilon $-accurate stationary policy only requiring one trajectory at each iteration. While the convergence results of MPO given in (Yang et al., 2019) are **incorrect** (Please see the openreview of **Paper613 AnonReviewer2** in https://openreview.net/forum?id=SkxpDT4YvS )，e.g., a key equality (40) at page 16 of (Yang et al., 2019) is not true.

We **then** compare our variance-reduced method (i.e., VR-BGPO) with the variance-reduced method (i.e., VRMPO )  of (Yang et al., 2019):

1) Our VR-BGPO uses a useful **momentum iteration** $ \theta_{k+1} = \theta_k + \eta_k(\tilde{\theta}_{k+1}-\theta_k)$ at the step 5 of our Algorithm 2. While VRMPO of (Yang et al., 2019) does **not** use a similar iteration.

2) Our VR-BGPO uses the variance-reduced technique of STORM without relying on large batch-size. While VRMPO of (Yang et al., 2019) uses the variance-reduced technique of SPIDER relying on large batch-size.
Clearly, the policy gradient estimator used in our VR-BGPO is **totally different** from the policy gradient estimator used in VRMPO of (Yang et al., 2019). At the same time, our VR-BGPO algorithm is a single-loop, while VRMPO algorithm is a double-loop.

3) We provide a solid convergence analysis for our VR-BGPO, and prove that our VR-BGPO has a sample complexity of $O(\epsilon^{-3})$ for finding an $\epsilon $-accurate stationary policy only requiring one trajectory at each iteration. Although  (Yang et al., 2019) also provided the convergence properties of VRMPO, the convergence results of  VRMPO given in (Yang et al., 2019) are **incorrect** (Please see the openreview of **Paper613 AnonReviewer2** in https://openreview.net/forum?id=SkxpDT4YvS )，e.g., the derivation in (55) at page 20 of (Yang et al., 2019) is not true.

---

> ### Author Response · Authors · 2021-11-12
> **Overall Responses (2)**
>
> Our methods (i.e., BGPO and VR-BGPO) are **very different** from the MBPG method of (Huang et al., 2020, http://proceedings.mlr.press/v119/huang20a/huang20a.pdf).
>
> 1)  Our BGPO and VR-BGPO use a useful **momentum iteration** $ \theta_{k+1} = \theta_k + \eta_k(\tilde{\theta}_{k+1}-\theta_k)$ at the step 5. While MBPG of (Huang et al., 2020) does **not** use a similar iteration.
>
> 2) Our BGPO and VR-BGPO use a **constant learning rate**(i.e., $\lambda$) at the step 4 of our Algorithm 1 and 2. While MBPG only uses a **decreasing learning rate** (i.e., $\eta_t = \frac{k}{(m+\sum_{i=1}^tG^2_i)^{1/3}}$ and $\eta_t = \frac{k}{(m+t)^{1/3}}$) at step 10 of Algorithms 1-3 in (Huang et al., 2020) .
>
> 3) Our BGPO and VR-BGPO consider the **constrained** policy problem with convex set $\Theta$. While MBPG of  (Huang et al., 2020) consider the **unconstrained** policy problem.
>
> 4) We study both the non-variance-reduced and variance-reduced policy gradient methods, i.e., BGPO and VR-BGPO. While  (Huang et al., 2020) only study the variance-reduced policy gradient methods, i.e., MBPGs. **NOTE THAT**: The basic momentum (non-variance-reduced) policy gradient estimator (the step 6 of Algorithm 1) used in our BGPO is **very different** from the momentum variance-reduced policy gradient estimator (the step 6 of Algorithm 2) used in our VR-BGPO requiring an importance sampling weight.
> To the best of our knowledge, our BGPO method is the first work to use the basic momentum policy gradient estimator **without** requiring an importance sampling weight in policy gradient methods.
>
>
> 5) Since our BGPO/VR-BGPO and MBPG of  (Huang et al., 2020)  algorithms are different, our convergence analysis are very different from  (Huang et al., 2020). E.g., In our convergence analysis, the mirror descent iteration is more complex than a simple gradient ascent iteration in (Huang et al., 2020) .

---

### Decision · Program_Chairs · 2022-01-20

**Decision:**

Accept (Poster)

**Comment:**

This paper proposes a policy gradient algorithm based on the Bregman divergence and momentum method. While one reviewer was initially concerned about the technical novelty of the paper given some existing works, after the author's response and paper revision, the reviewers are all convinced and have reached a consensus to accept this paper. Thus I recommend acceptance.